

# A non-stationary stochastic ensemble generator for radar rainfall fields based on the Short-Space Fourier Transform

Daniele Nerini[1,2], Nikola Besic[1,3], Ioannis Sideris[1], Urs Germann[1], Loris Foresti[1]

[1] Radar, Satellite and Nowcasting Division, MeteoSwiss, Locarno-Monti, Switzerland
[2] Institute for Atmospheric and Climate Science, ETH, Zurich, Switzerland
[3] Environmental Remote Sensing Laboratory, EPFL, Lausanne, Switzerland

*Correspondence to*: Daniele Nerini (Daniele.Nerini@meteoswiss.ch)

**Abstract.** In this paper we present a non-stationary stochastic generator for radar rainfall fields based on the
Short-Space Fourier Transform (SSFT). The statistical properties of rainfall fields often exhibit significant
spatial heterogeneity due to differences in the involved physical processes and influence of orographic forcing.
The traditional approach to simulate stochastic rainfall fields based on the Fourier filtering of white noise, also
known as fractional Brownian noise integration, is only able to reproduce the global power spectrum and spatial
autocorrelation of the precipitation fields. Conceptually similar to wavelet analysis, the SSFT is a simple and
effective extension of the Fourier transform developed for space-frequency localisation, which allows using
windows to better capture the local statistical structure of rainfall. The SSFT is used to generate stochastic noise
and precipitation fields that replicate the local spatial correlation structure, i.e. anisotropy and correlation range,
of the observed radar rainfall fields. The potential of the stochastic generator is demonstrated using four
precipitation cases observed by the 4[th] generation of Swiss weather radars that display significant non-
stationarity due to the coexistence of stratiform and convective precipitation, differential rotation of the weather
system and locally varying anisotropy. The generator is verified in its ability to reproduce both the global and the
local Fourier power spectra of the precipitation field. The SSFT-based stochastic generator can be applied and
extended to improve the probabilistic nowcasting of precipitation, design storm simulation, stochastic NWP
downscaling and also for other geophysical applications involving the simulation of complex non-stationary
fields.

**Keywords.** Precipitation, uncertainty, stochastic noise, non-stationarity, local Fourier transform, local
anisotropy, complex orography, nowcasting, design storms, NWP downscaling.



## 1. Introduction

Precipitation exhibits a large spatial and temporal variability over a wide range of scales (Lovejoy and Schertzer, 2006). Such complexity represents a continuous challenge in the practice of observing, understanding and simulating precipitation processes. Despite great efforts, the measurement and prediction of rainfall at fine

space-time resolutions is still associated with large errors. Consequently, it is of crucial importance to accurately characterise the uncertainty related to its measurement and prediction, in particular for hydrometeorological applications that are sensitive to input precipitation uncertainty (Zappa *et al.*, 2010).

For medium-range weather forecasting at synoptic and mesoscales, the forecast uncertainty is usually considered by perturbing the initial conditions of Numerical Weather Prediction (NWP) models to generate an ensemble of

possible future realisations. However, the NWP approach for uncertainty quantification is often not adapted for several hydrological applications, when high spatial and temporal resolution is required. A typical application for very-short term forecasting (nowcasting) of radar rainfall at 5 min update rate would need ideally the generation of 50-100 realisations of precipitation fields for the next 6 hours in less than 5 minutes. Assuming one second of computational time per realisation it already requires 60 minutes on a single processor (50 realisations x 72 lead

times x 1 second = 60 min). These requirements cannot be met by current operational NWP models and a different approach needs to be taken.

It has been long observed that rainfall fields exhibit spatial and temporal organization which is consistent with the scaling and multifractal framework (Schertzer and Lovejoy, 1987; Menabde *et al.*, 1997). This observation is convenient since it allows developing parsimonious stochastic simulation methods which are able to generate

realistic precipitation fields that depend only on a few parameters. Stochastic rainfall generators are effective in characterizing the uncertainty related to the measurement and forecasting of precipitation and they constitute the core of most statistical design storm techniques, probabilistic nowcasting and NWP downscaling methods.

An ideal stochastic rainfall generator for hydrometeorological applications should be able to:

1.  Simulate large precipitation fields (> 500x500 grid points) with minimum computational time (< 1-10
seconds).

2.  Reproduce the observed spatial and temporal autocorrelation of the observed precipitation fields.

3.  Reproduce the observed power law scaling behaviour of the rainfall fields (ideally the multifractal scaling of different moments of order *q*).

4.  Reproduce the observed marginal distribution of rainfall rates (often skewed).

5.  Take into account the rainfall intermittency and the effects due to the rain / no-rain transition.

6.  Integrate in a flexible way possible sources of predictability (NWP forecasts, analogues, orography, etc).

7.  Handle the spatial and temporal *non-stationarity* of the statistical properties of precipitation.

This paper focuses particularly on point 7 and presents a non-stationary spatial stochastic rainfall generator

which is able to reproduce the local variability and anisotropy of the observed precipitation fields.

### 1.1 Brief review of spatial stochastic rainfall generators

There is abundant literature on random field generators using geostatistical (Goovaerts, 1997; Lantuéjoul, 2002) and multifractal approaches (Menabde *et al.*, 1997; Lovejoy and Schertzer, 2006). A widely used covariance-



based geostatistical random field generator in hydrology is based on the turning bands method (TBM, see a review in Lantuéjoul, 2002), which reduces the problem of simulating a multidimensional Gaussian random field to the simulation of a set of independent unidimensional autocorrelated Gaussian processes along random orientations (for practical implementations in hydrology refer to Mantoglou and Wilson, 1982; Tompson *et al.*,

1989). In the context of precipitation, Leblois and Creutin (2013) adapted the TBM to simulate unconditional stochastic fields which reproduce the correct intermittency and advection of precipitation fields, which was further extended for ensemble precipitation nowcasting by Caseri *et al.* (2016). Schleiss *et al.* (2014) also followed the geostatistical approach by using Sequential Gaussian Simulations (SGS) to generate conditional and unconditional radar rainfall fields which realistically decay towards zero when moving out from the wet regions

(concept of "dry drift"). Despite the speed up strategies implemented, both the SGS and TBM approaches are still too computationally demanding to generate large precipitation ensembles over extended domains for real-time applications.

Another class of algorithms relies on the Fast Fourier Transform (FFT), which has efficient and easy to use libraries in most programming languages. The Wiener-Khintchine theorem (Wiener, 1930; Khintchine, 1934)

states that the Power Spectral Density (PSD, generally referred to as "power spectrum") of a wide sense stationary random process can be obtained as the Fourier transform of the autocorrelation (autocovariance) function of that process. Thus, it becomes clear that stochastic simulation methods based on covariance or variogram can be easily reformulated by exploiting the FFT to achieve increased computational efficiency (Marcotte, 1996). In addition, the representation of data in Fourier space can enhance our understanding of the

underlying generating mechanism and scaling behaviour of rainfall.

One of the most straightforward and widely used stochastic noise generation techniques is the filtering of white noise field using a Fourier spectrum characterised by a power law behaviour (linear relationship in log-log plot of power vs. frequency). The inverse Fourier transform of the filtered noise field directly provides a correlated Gaussian field which reproduces the chosen power law spectrum (for example corresponding to an exponential

covariance function). This stochastic generator will be referred to as global FFT-noise throughout the paper. The FFT provides an efficient solution to simulate fractional Brownian motion fields (Mandelbrot and Van Ness, 1968), multidimensional and also multivariate autocorrelated random fields (Robin *et al.*, 1993). Fractional noise integration is particularly useful to model geophysical processes which exhibit Fourier spectra following power laws of fractional order (Schertzer and Lovejoy, 1987), in particular precipitation fields (Menabde *et al.*, 1997;

Lovejoy and Schertzer, 2006).

There is a long list of probabilistic precipitation estimation, nowcasting, design storm and NWP downscaling techniques that exploit the FFT-noise to generate ensembles of stochastic rainfall fields. One of the most advanced probabilistic nowcasting techniques is the Short-Term Ensemble Prediction System (STEPS, Bowler *et al.*, 2006), which represents the uncertainty due to rainfall growth and decay processes by adding stochastic

perturbations to a deterministic radar extrapolation forecast. The radar rainfall field is decomposed in Fourier space into an 8-level multiplicative cascade to allow representing the scale-dependence of the predictability of precipitation. The rainfall cascade is blended scale by scale with a cascade of spatially correlated noise fields to consider the different rate of development of precipitation at different spatial scales. STEPS is operationally used at the Australian Bureau of Meteorology, the UK MetOffice and the Royal Meteorological Institute of Belgium

and is in continuous development (see Seed *et al.*, 2013; Foresti *et al.*, 2016). The STEPS framework was also used for design storm generation to produce realistic synthetic rainfall fields (Seed *et al.*, 1999; Niemi *et al.*,




2016). SBMcast is another well-known probabilistic nowcasting system which employs the FFT-noise technique (Berenguer *et al.*, 2011). It is based on the "String of Beads" model of Pegram and Clothier (2001a,b), which simulates the joint evolution of global radar rainfall fields statistics that define the general storm behaviour (wet area ratio and image mean flux, i.e. rainfall fraction and mean rainfall) and the local rainfall. Metta *et al.* (2009)

developed a probabilistic nowcasting system which exploits a Langevin-type model for the evolution of Fourier phases to generate isotropic stochastic rainfall fields. A similar approach was implemented within the RainFARM model of Rebora *et al.* (2006b) for stochastic NWP precipitation downscaling in space and time. STREAP (Paschalis *et al.*, 2013) also uses the Fourier transform for design storm generation to produce Gaussian random fields with an exponentially decaying isotropic autocorrelation function. Atencia and Zawadzki

(2014) use the power spectrum of the last observed radar rainfall field to generate spatially correlated and anisotropic stochastic perturbations for ensemble precipitation nowcasting. Niemi *et al.* (2014) parametrised the 2D power spectrum of precipitation fields to obtain a Fourier filter able to simulate anisotropic stochastic rainfall fields for design storm generation (Niemi *et al.*, 2016).

Stochastic noise generators are also increasingly used to characterise the residual radar measurement uncertainty

for ensemble quantitative precipitation estimation (ensemble QPE). An important group of techniques generates the stochastic perturbations by LU decomposition of the covariance matrix which represents the multiplicative radar and rain gauge errors (see e.g. Ciach *et al.*, 2007; Germann *et al.*, 2009; Villarini *et al.*, 2009). Given the direct link between the covariance and the Fourier spectrum via the Wiener-Khintchine theorem, it is not a surprise to observe that FFT-based noise generators are also used for ensemble radar-based QPE (Jordan *et al.*,

2003; Pegram *et al.*, 2011). An interesting question is how to condition the unconditional rainfall ensembles to rain gauges, e.g. as done in the conditional merging of Sinclair and Pegram (2005) or the random mixing approach of Bardossy and Hörning (2016). It is worth mentioning that Velasco-Forero *et al.* (2009) and Schiemann *et al.* (2011) also took advantage of the Wiener-Khintchine theorem to obtain 2D positive definite correlograms directly from radar rainfall images for improved blending with rain gauge measurements.

**1.2 Limitation of current stochastic rainfall generators**

A major limitation and concern of all the cited stochastic generators is that they assume spatial stationarity, i.e. uniformity of the generator across space. As a consequence the generator is only able to produce spatially homogeneous noise fields (see point 7 in the requirements of the generator at the beginning of the introduction). More precisely, they cannot represent and simulate non-stationary fields that display an heterogeneous

distribution of autocorrelation range and/or anisotropy.

The non-stationarity in the local statistical properties becomes easily visible when stratiform and convective precipitation coexist in different regions of the radar domain or when there are clear local anisotropies in different directions (see for example Fig.. 1). The need for a non-stationary stochastic noise generator is even more relevant in regions with complex orography, such as Switzerland, where the statistical properties and

predictability of precipitation are strongly controlled by orographic forcing (Panziera and Germann, 2010; Mandapaka *et al.* 2012; Foresti and Seed, 2014; Foresti and Seed, 2015). A few approaches have been proposed to adapt the stochastic generators to better capture the local variability of precipitation. Pathirana and Herath (2002) and Badas *et al.* (2006) proposed to filter or remove the heterogeneities of the precipitation field before applying a homogeneous multifractal downscaling technique and re-adding the trend after running the stochastic

simulation. A similar strategy was employed in a recent STEPS implementation, which computes the local mean





and variance of the rainfall field on a regular grid to remove the local non-stationarity before adding the homogeneous stochastic perturbations (Seed *et al.*, 2013). However, these strategies are not able to generate noise fields that reflect the local 2D power spectrum and anisotropy of precipitation fields. When applied for nowcasting of weather situations presenting strong local anisotropies, the stochastic perturbations gradually

destroy the original spatial structure and important information is rapidly lost. The need for a more flexible approach to directly consider the non-stationarity of real-world geophysical fields such as rainfall was already pointed out by Lovejoy and Schertzer (2013). The same authors argue that the estimation of the generator over small sections of the field in a moving window fashion suffers from the small sample size. Therefore, they propose to model the spatial variation of the variables defining the generator using a nonlinear extension of

Generalized Scale Invariance, whose estimation, however, is also not trivial (Lovejoy and Schertzer, 2013).

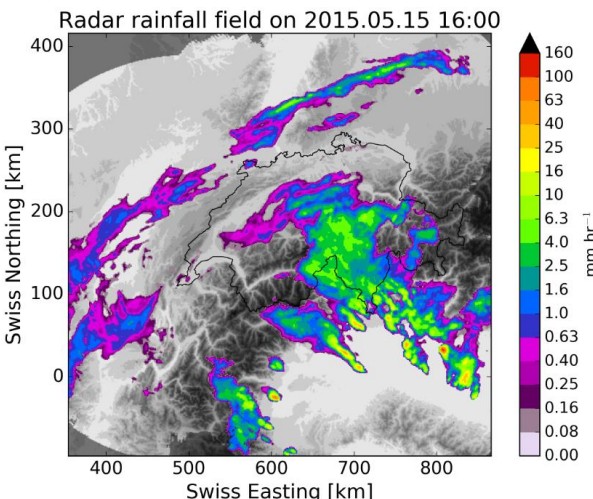

**Figure 1.** Example of non-stationarities in a radar rainfall field of 512x512 km centred over Switzerland. Two distinct anisotropies coexist in the same image, while convection develops only to the south of the Alps.

### 1.3  Contribution of this study

The contribution of this paper is the design of a non-stationary stochastic rainfall generator based on the Short-Space Fourier Transform (SSFT, Hinman *et al.*, 1984). The idea is to extend the Short-Time Fourier Transform (STFT, Kovačević *et al.*, 2013) for two dimensional spatial fields both for local Fourier analysis and stochastic simulation. Conceptually similar to the wavelet transform (Kumar and Foufoula-Georgiou, 1997; Kovačević *et al.*, 2013), the STFT and SSFT allow time- and space-frequency localisation respectively, i.e. they enable

performing a local moving window Fourier analysis to account for non-stationarity in the statistical properties of the signal (in our case a 2D precipitation field). The traditional global FFT-noise technique can be easily extended by performing local Fourier filtering of white noise in a moving window fashion using as filter the local Fourier transform of the precipitation field. The obtained locally correlated noise field (denoted as SSFT-noise) automatically reproduces the local correlation structure of the target precipitation field. In this study we

want to investigate to which extent it is reasonable to estimate a local stochastic generator from the data considering the larger uncertainty due to the reduced sample size.

In order to demonstrate the advantage of using a non-stationary noise generator we selected four non-stationary rainfall fields from the 4[th] generation of dual-pol Swiss weather radar composite. We verified the ability of the





generator in replicating the global and local statistical properties of the precipitation fields (Fourier spectrum, anisotropy). The SSFT-noise generator will open the doors for enhanced probabilistic precipitation estimation and nowcasting, design storm generation and stochastic NWP downscaling. Geostatistical interpolation methods considering the local anisotropy (e.g. Boisvert *et al.*, 2009; Gyasi-Agyei, 2016) could also be extended to

perform local stochastic simulation.

The paper is organized as follows. Section 2 describes the global Fourier-based method for generating spatially correlated stochastic noise fields (FFT-noise). Section 3 explains the generation of locally correlated stochastic noise based on the Short-Space Fourier Transform (SSFT-noise) and Section 4 makes use of synthetic data produced with parametric methods in order to illustrate and test the functioning of the novel approach. Section 5

presents four selected radar rainfall fields characterised by strong non-stationarities. The four precipitation cases are thus used for the verification of the local SSFT-noise generator in comparison with the global FFT-noise generator, in particular its ability to reproduce both the global and local structure of precipitation fields. At the end of the Section 5, we provide two examples demonstrating the potential applications of the local noise generator. First, we present stochastic simulations of synthetic radar rainfall fields by appropriate local

adjustment of the noise fields. Then, we introduce the temporal structure of the precipitation field by using the local noise as shock term in an autoregressive process in the context of a simplified stochastic precipitation nowcast. Section 6 concludes the paper and enumerates possible applications and extensions of the generator for stochastic precipitation simulation.

## 2.  Stationary stochastic generator

This section formulates the model to generate two-dimensional spatially correlated noise and precipitation fields. Such fields should reproduce the spatial scaling, and thus spatial autocorrelation, of the target radar rainfall field. The numerical simulation of such random fields is based on the idea of filtering a Gaussian white noise field using the power law Fourier spectrum of the precipitation field, also known as fractional integration (Schertzer and Lovejoy, 1987). The following sections explain the basic principles of the Fourier transform and how it can

be used for simulation of stationary spatially correlated random fields.

### 2.1  Fourier Transform

The corner stone of signal processing theory (Stankovic Lj. *et al.*, 2013) is the idea that a signal $x(t)$ can be represented as an infinite weighted sum of harmonic sine waves of infinitesimally close frequencies $f$:

$$x(t) = \int_{-\infty}^{+\infty} X(f)e^{j2\pi ft}df, \qquad \text{Eq. 2.1}$$

where $X(f)$ represents the complex Fourier frequency spectrum. It contains both the amplitude ($|X(f)|$) and the

phase ($\varphi$) information ($X(f) = |X(f)|e^{j\varphi}$). The spectrum, obtained by means of the Fourier transform (FT), is actually the same signal, only represented in the frequency domain:

$$X(f) = \mathcal{F}\{x(t)\} = \int_{-\infty}^{+\infty} x(t)e^{-j2\pi ft}dt, \qquad \text{Eq. 2.2}$$

where $\mathcal{F}\{x(t)\}$ denotes the Fourier transform of the signal $x(t)$. Therefore, the Fourier transform could be essentially interpreted as the transformation of a signal from a given space to the space of frequencies, where the




energy/average power of a signal is conserved (Parseval's theorem). Signal $x(t)$ and its Fourier spectrum $X(f)$ are often called the FT pair.

If the signal $x(t)$ is deterministic (or transient stochastic), its energy spectral density can be defined as $S(f) = |X(f)|^2$, while in the case of a non-transient stochastic signal we ought to define its power spectral density,

being $S(f) = \lim_{T \to \infty} \frac{|X(f)|^2}{2T}$. Both the energy spectral density and the power spectral density form the Fourier transformation pair with the autocorrelation function of a corresponding signal:

$$R(\tau) = \int\limits_{-\infty}^{+\infty} x(t)x(t+\tau)dt = \mathcal{F}^{-1}\{S(f)\}. \qquad \textbf{Eq. 2.3}$$

In the latter case, of non-transient stochastic signals, this relation is called Wiener-Khintchine theorem, and it is very relevant in the analysis of stochastic fields. In other words, the autocorrelation function of a signal can be simply obtained as the inverse FT of its power spectral density.

That leads us to the next step, which is the expansion of the Fourier transform to the analysis of stochastic spatial signals – stochastic fields. The expanded, two-dimensional Fourier transform (2D FT) would be, as a matter of fact, the corner stone of image processing theory (Stankovic S. *et al.*, 2012). Namely, using 2D FT a field $f(x,y)$ can be transformed from a two-dimensional space to the two-dimensional frequency domain:

$$F(f_1,f_2) = \mathcal{F}\{f(x,y)\} = \iint\limits_{-\infty}^{+\infty} f(x,y)\,e^{-j2\pi(f_1x+f_2y)}dxdy, \qquad \textbf{Eq. 2.4}$$

with $f_1 = \frac{1}{x}$ and $f_2 = \frac{1}{y}$ being the spatial frequencies in $x$ and $y$ directions. Analogously to the one-dimensional

signal, which can be represented as a weighted sum of one-dimensional sinusoids, a field is represented by a weighted sum of sinusoids that vary in the $x - y$ plane:

$$f(x,y) = \mathcal{F}^{-1}\{F(f_1,f_2)\} = \iint\limits_{-\infty}^{+\infty} F(f_1,f_2)\,e^{j2\pi(f_1x+f_2y)}df_1df_2, \qquad \textbf{Eq. 2.5}$$

with the complex spectrum $F(f_1,f_2) = |F(f_1,f_2)|e^{j\varphi}$. The power spectral density of a stochastic field is estimated on $\Delta x \times \Delta y$ portion of a plane as $S(f_1,f_2) = \lim_{\Delta x \Delta y \to \infty} \frac{|F(f_1,f_2)|^2}{\Delta x \Delta y}$. Again, analogously to the 1D case, it forms the Fourier transformation pair with the two-dimensional autocorrelation function (Wiener-Khintchine

theorem):

$$R(\xi,\zeta) = \iint\limits_{-\infty}^{+\infty} f(x,y)f(x+\xi,y+\zeta)\,dxdy = \mathcal{F}^{-1}\{S(f_1,f_2)\}. \qquad \textbf{Eq. 2.6}$$

In order to obtain the autocorrelation function, the signal needs to be standardised by removal of the mean and division by the standard deviation, otherwise only the non-centred autocovariance is obtained. Using the Wiener-Khintchine theorem and exploiting the speed of FFT one can compute the autocorrelation functions of regularly sampled signals very efficiently (Marcotte, 1996; Velasco-Forero *et al.*, 2009; Schiemann *et al.*, 2011). On the

other side, the classical approach to derive the spatial autocorrelation function would require computing all the Euclidean distances between pairs of points located at different ranges and along different directions.

Examples of two-dimensional power spectral densities and autocorrelation functions, for the case studies considered in this paper, are given in Fig. 8 and Fig. 9. Aside from the complete two-dimensional information about field power spectral density ($S(f_1,f_2)$), it is common to use the one-dimensional, simplified representation

($S_{1D}(f)$), obtained by radially averaging $S(f_1,f_2)$ around the $S(0,0)$ point. This allows us to analyse the scaling



behaviour of rainfall, by characterising the power spectral curve with the spectral exponent $\beta$, obtained by its power law approximation in the log-log plot $S_{1D}^a(f) = \frac{1}{f^\beta}$.

One of the most noteworthy properties of the Fourier transform is a significantly facilitated convolution in the spatial (or temporal) domain. Namely, the convolution of two fields $f_1(x, y)$ and $f_2(x, y)$ forms the FT pair with

the product of their respective Fourier transforms $F_1(x, y)$ and $F_2(x, y)$:

$$\mathcal{F}\{f_1(x, y) * f_2(x, y)\} = F_1(f_1, f_2)F_2(f_1, f_2). \qquad \textbf{Eq. 2.7}$$

### 2.2 Global filtering of white noise with FFT

In analysing real, observed fields, we actually never deal with the continuous functions (e.g. $r(x, y)$) but rather with the discrete ones. That would mean that a two dimensional field can be represented by a matrix of finite size ($\mathbf{r}_{M \times N}$), with the same being true for its Fourier pair $\mathbf{R}_{M \times N}$. In this case, the transformation from Eq. 2.4 takes the

form of a Discrete Fourier Transform (DFT):

$$\mathbf{R}(m, n) = \frac{1}{\sqrt{MN}} \sum_{l=0}^{N-1} \sum_{k=0}^{M-1} \mathbf{r}(k, l) e^{-j2\pi\left(\frac{km}{M} + \frac{ln}{N}\right)}, \qquad \textbf{Eq. 2.8}$$

which, particularly if $M$ and $N$ are powers of 2 or have only small prime factors, can be efficiently implemented using Fast Fourier Transform (FFT) algorithms (Cooley and Tukey, 1965). The transformation preserves all the properties presented in the preceding section, out of which the facilitated convolution especially proves to be useful in generating the stationary correlated noise.

Namely, the straightforward way of generating a stationary correlated noise field, i.e. a stochastic field with the overall spatial correlation of a natural rainfall field, is the non-parametric filtering of a white noise field ($\mathbf{n}$). This basically assumes convolving a generated white Gaussian noise field, characterised with zero mean and unity variance, with the selected rainfall field. If we employ the FFT algorithm to calculate the DFT of a rainfall field ($\mathbf{R}$), and the DFT of a noise field ($\mathbf{N}$), a stationary correlated noise field ($\mathbf{n}_{scr}$) can be derived as:

$$\mathbf{n}_{scr} = \text{FFT}^{-1}\{\mathbf{R} \circ \mathbf{N}\}, \qquad \textbf{Eq. 2.9}$$

with $[\cdot] \circ [\cdot]$ being the pointwise (Hadamard) product. Given the quasi constant power spectral density of white noise, this way we mostly alter the spectral phase of a rainfall field, without significantly modifying the amplitude. It makes the resultant spectrum of one realisation of correlated noise field ($\mathbf{N}_{scr}$) almost identical to the spectrum of the considered rainfall field ($\mathbf{R}$), whose inverse FFT corresponds to the spatial autocorrelation function. The latter could also be achieved by generating a noise field which can be used to directly perturb the

phase of $\mathbf{R}$ (Metta *et al.*, 2009). In this case we can avoid calculating the Fourier transform of the noise, FFT$\{\mathbf{n}\}$, which can slightly contribute to the computational efficiency. Alternatively, the rainfall field spectrum ($\mathbf{R}$) could be as well approximated by a suitable power spectral law, resulting in the parametric filtering of white noise field.

The most important drawback of the obtained stationary correlated noise field is its spatial stationarity. Namely,

as already pointed out, the obtained spectrum ($\mathbf{N}_{scr}$) corresponds to the one of the selected rainfall field ($\mathbf{R}$), which implies that the two fields have the same global field anisotropy properties and scaling behaviour. This however does not suggest that a correlated noise field exhibits the same local behaviour as its rainfall field counterpart, i.e. two fields do not have the same local field anisotropy properties.



### 2.3 Non-parametric and parametric filters

In Sect. 2.2 the white noise field was filtered by the actual DFT of the rainfall field without any parametrisation of the distribution of power as a function of frequency and direction. This is referred to as non-parametric filtering of white noise and allows automatic simulation of the observed anisotropies and variability of the
spectral slope as a function of scale (Seed *et al.*, 2013). This approach is particularly effective for cases with sufficient precipitation coverage over the radar domain and in the absence of measurement noise.

On the other hand, for cases with a small fraction of precipitation, the Fourier transform suffers from the small sample size. As a consequence, the power spectrum becomes noisy and it may be hard to observe the anisotropy or the power law rainfall scaling behaviour. In order to increase the sample size and reduce the noise, a more
stable Fourier power spectrum can be computed by averaging the spectra of a sequence of precipitation fields as done by Niemi *et al.* (2014). This strategy could be helpful in particular when computing local Fourier transforms for the SSFT (see Sect. 3.1).

Instead, it is common to parametrize the 2D or 1D power spectra using simple power law scaling models for rainfall. The parametrisation of 2D power spectra can be achieved by constructing an anisotropic filter that
considers isolines of constant power as a function of frequency based on the principle of Generalized Scale Invariance (Niemi *et al.*, 2014). A simpler isotropic scaling model can be constructed from the radially averaged 1D power spectrum by neglecting the field anisotropy. A power law fit of the rainfall field power vs. spatial frequency in log-log plot is generally sufficient for the task. More refined models acknowledge the presence of a scaling break in the power law around 10-20 km by fitting two spectral exponents ($\beta_1$ and $\beta_2$), which represent
the different scaling regimes of the large scale precipitation features and the small scale convective cells respectively (Gires *et al.*, 2011; Seed *et al.*, 2013).

### 3.   Non-stationary stochastic generator

The following section introduces the concept of Short-Space Fourier Transform, and explains how the former is used in overcoming the evoked issue of spatial stationarity in the simulation of spatially correlated noise fields.

### 3.1 Short-Space Fourier Transform

Although the phase of the Fourier spectrum contains an information about the temporal distribution of spectral components, its interpretation proves to be quite difficult. Therefore, due to the increasing need to face the non-stationarity of signal, numerous advanced methods have been proposed (Stankovic Lj. *et al.*, 2013), all of which can be gathered under the umbrella of time-frequency signal analysis (TFSA).

The most intuitive and representative TFSA method is the Short-Time Fourier Transform (STFT), which provides Fourier spectra of localised parts of a signal:

$$X^{ST}(t,f) = \int_{-\infty}^{+\infty} x(t+\tau)w(\tau)e^{-j2\pi f\tau}d\tau, \qquad \textbf{Eq. 3.1}$$

where $w(\tau)$ is the window localising the particular part of a signal around the moment $t$ $(x(t+\tau)w(\tau))$. The introduced properties of FT (power spectral density, convolution, auto-correlation), are valid for the obtained spectrum $X^{ST}(t,f)$, in the same way they are for the spectrum of the entire signal.

The original signal can be obtained by integrating all individual inverse FT:




$$x(\tau) = \int\limits_{-\infty}^{+\infty} x(t+\tau)w(\tau)\,dt = \int\limits_{-\infty}^{+\infty}\int\limits_{-\infty}^{+\infty} X^{ST}(t,f)\,e^{j2\pi f\tau}\,df\,dt, \qquad \text{Eq. 3.2}$$

which, due to the overlapping of segments, provides a better reconstruction, particularly important in case of highly non-stationary signals. If $\int_{-\infty}^{+\infty} w(\tau) = 1$, despite the overlapping, there is no need to normalize the obtained result. Theoretically, it would be enough to perform the inverse FT at the time step $\Delta t = 2T$, which corresponds to the width of the window.

Analogously to the Fourier Transform, the Short-Time Fourier Transform can be expanded to the spatial domain, becoming the Short-Space Fourier Transform (SSFT):

$$F^{SS}(x,y,f_1,f_2) = \int\limits_{-\infty}^{+\infty}\int\limits_{-\infty}^{+\infty} f(x+\epsilon, y+\mu)w(\epsilon,\mu)e^{-j2\pi(f_1\epsilon+f_2\mu)}d\epsilon\,d\mu, \qquad \text{Eq. 3.3}$$

with the window $w(\epsilon,\mu)$ now localising the particular segment of a field around the position $x,y$ ($f(x+\epsilon, y+\mu)w(\epsilon,\mu)$). The original field is obtained as:

$$f(\epsilon,\mu) = \int\limits_{-\infty}^{+\infty}\int\limits_{-\infty}^{+\infty} f(x+\epsilon, y+\mu)w(\epsilon,\mu)dx\,dy$$

$$= \int\limits_{-\infty}^{+\infty}\int\limits_{-\infty}^{+\infty}\int\limits_{-\infty}^{+\infty}\int\limits_{-\infty}^{+\infty} F^{SS}(x,y,f_1,f_2)e^{j2\pi(f_1\epsilon+f_2\mu)}df_1\,df_2\,dx\,dy. \qquad \text{Eq. 3.4}$$

The critical part concerns choosing the size of the window, as well as its shape. Namely, due to the uncertainty
principle, a smaller window in the space domain, i.e. a better space resolution, results in a worse frequency resolution, and vice versa. Choosing an appropriate function $w(\epsilon,\mu)$ can decrease the otherwise inevitable uncertainty.

### 3.2 Window functions

Equivalently to the convolution in space domain, which corresponds to the product in its frequency counterpart
(Eq. 2.7), the product in space domain forms a Fourier pair with the convolution of the spectra. Therefore, the localisation of the field in space, with the window function $w(\epsilon,\mu)$, implies the localisation in frequency through the convolution with the FT of the window function $W(f_1,f_2)$.

The most intuitive choice of a window would be a simple rectangular function of width $2T$ (Fig. 2), which in case of a field analysis, through the product of two separable functions, becomes a square:

$$w_R(\epsilon,\mu) = w_R(\epsilon)w_R(\mu) = \begin{cases} 1 & \text{for } |\epsilon| < T \text{ and } |\mu| < T \\ 0 & \text{elsewhere} \end{cases}, \qquad \text{Eq. 3.5}$$

and whose FT is as well a product of the FTs of two separable functions:

$$W_R(f_1,f_2) = \mathcal{F}\{w_R(\epsilon)\}\mathcal{F}\{w_R(\mu)\} = \frac{\sin(2\pi f_1 T)\sin(2\pi f_2 T)}{\pi^2 f_1 f_2}. \qquad \text{Eq. 3.6}$$

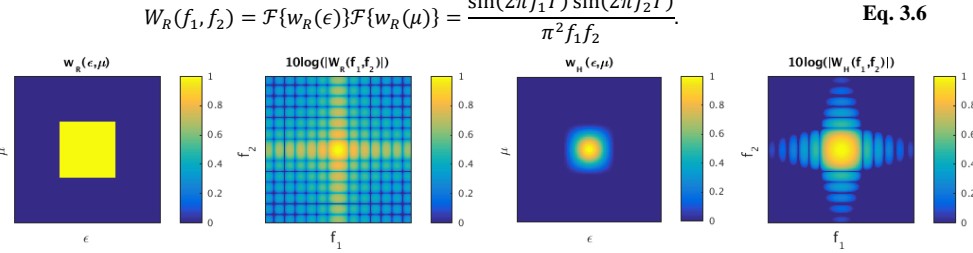

**Figure 2.** FT pairs for rectangular and Hanning windows (values normalised to unity).





However, as it can be seen in Fig. 2, the side lobes of $W_R(f_1, f_2)$ are somewhat strong, causing the poor localisation in the frequency domain. Hence the need to come up with a more suitable form of a window, which can reduce the overall uncertainty, through the good localisation in both space and frequency domain. Although the best choice in terms of uncertainty would be the Gaussian window, due to the performance in the context of our application, we have adopted the Hanning window:

$$w_H(\epsilon, \mu) = \begin{cases} \frac{1}{4}\left(1 + \cos\left(\frac{\epsilon\pi}{T}\right)\right)\left(1 + \cos\left(\frac{\mu\pi}{T}\right)\right) & \text{for } |\epsilon| < T \text{ and } |\mu| < T, \\ 0 & \text{elsewhere} \end{cases}$$ 

**Eq. 3.7**

The corresponding FT decreases proportionally to the $f^3$,

$$W_H(f_1, f_2) = \frac{\pi^2 \sin(2\pi f_1 T)\sin(2\pi f_2 T)}{4\pi^2 f_1 f_2(\pi^2 - 4\pi^2 f_1^2 T^2)(\pi^2 - 4\pi^2 f_2^2 T^2)},$$

**Eq. 3.8**

leading to the better localisation in the frequency domain with respect to the rectangular window (see Fig. 2).

### 3.3 Local filtering of white noise with SSFT

As already stated previously, in the context of numerous diverse applications, the most obvious drawback of a conventionally derived correlated noise field is its spatial stationarity. Tackling that issue, which is the main motivation and idea of this work, is conceptually quite simple: it comes down to the replacement of the Fourier transform, in both non-parametric and parametric filters, with the Short-Space Fourier transform.

Namely, instead of deriving either a parametrised or non-parametrised (Eq. 2.8) Fourier spectrum of the entire rainfall field in order to filter white noise, here we derive a number of local spectra by focusing at different parts of the field. This is done by sliding and cantering the window at positions $(p_1, p_2)$:

$$\mathbf{R}(p_1, p_2, m, n) = \frac{1}{\sqrt{MN}} \sum_{l=0}^{N-1} \sum_{k=0}^{M-1} \mathbf{r}(p_1 + k, p_2 + l)\mathbf{w}(k, l)e^{-j2\pi\left(\frac{km}{M} + \frac{ln}{N}\right)},$$

**Eq. 3.9**

where $p_1$ varies in the range $\left(1, \frac{N}{\Delta}\right)$ (equivalent for $p_2$), with $2T \times 2T$ being the size of the non-zero portion of the window function. The maximum $\Delta$ value corresponds to the size of the window ($2T$) to avoid gaps, yet the overlapping ($\Delta < 2T$) can ensure a smoother representation of the non-stationarity. Although the integration could be performed only in the non-zero part of the matrix $\mathbf{w}$, the window is zero padded to the size of the field $\mathbf{r}_{M \times N}$, resulting in a better frequency sampling resolution.

In the following step, the local spectra are convolved with a white noise field $\mathbf{n}$, producing the local spatially correlated noise:

$$\mathbf{n}_{scr}(p_1, p_2) = \text{FFT}^{-1}\{|\mathbf{R}(p_1, p_2)| \circ \mathbf{N}\}.$$

**Eq. 3.10**

By using only the amplitude spectrum of the rainfall field (its absolute value), rather than perturbing the phase of the rainfall field, we are actually imposing the random phase of the white noise field, and slightly perturbing the amplitude. This way we preserve the local anisotropy, while simultaneously benefiting from the phase coherence in the final recomposition, i.e. it decreases the need for the overlapping. The recomposition of the final non-stationary field is obtained by summing over all the local spatially correlated noise fields:

$$\mathbf{n}_{scr} = \sum_{p_1=1}^{\frac{N}{\Delta}} \sum_{p_2=1}^{\frac{M}{\Delta}} \mathbf{n}_{scr}(p_1, p_2).$$

**Eq. 3.11**





where $\frac{N}{\Delta}$ is the number of windows in the $k$ direction and $\frac{M}{\Delta}$ is the number of windows in the $l$ dimension. In other words, if the rainfall field $\mathbf{r}_{M \times N}$ is represented by a set of 10x10 local FFTs (without overlapping), one must sum over the 100 correlated noise fields, which are weighted by the window for spatial localisation of the noise.

**4.    Synthetic example**

The novel concept can be intuitively demonstrated, or rather illustrated, using a synthetic dataset. Namely, instead of using Eq. 2.9 (i.e. the non-parametric approach), here we rely on parametric filtering in order to prescribe a power spectrum to a field of Gaussian white noise. The most straightforward approach is to use an isotropic power law filter:

$$H(k) = k^{-\beta/2},$$    **Eq. 4.1**

where $H(k)$ is the parametric filter as a function of $k$, the spatial frequency or wave number. It follows a stationary, isotropic field of correlated noise whose spectral slope is nearly equal to $\beta$. It can be seen in Fig. 3 that $\beta$ controls the smoothness of field, or autocorrelation range, as a more negative $\beta$ results in a more pronounced spectral slope and consequently in a higher power at larger wavelengths.

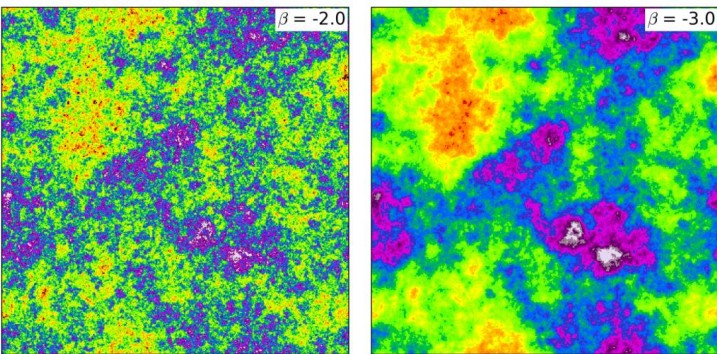

**Figure 3.** Examples of 512x512 synthetic fields of correlated noise produced with an isotropic power law filtering of Gaussian with noise using two different values of $\beta$. Both noise fields share the same random seed.

The same filtering can be applied locally, so that a set of $\beta$ is used to control the autocorrelation range within successive windows over the same field of Gaussian white noise. Figure 4a presents a set of non-stationary autocorrelation functions in terms of autocorrelation range overlaid on the resulting correlated noise. In other

words, these parametric 2D autocorrelation functions are used to filter white noise, locally, using a set of windows without overlapping. This image represents our target field. It is easy to notice that the autocorrelation range of the background noise field decreases when going from the top left to the bottom right corner of the image. In the following experiment we compute the global and local Fourier transforms of the target noise field in order to filter the same white noise field but with non-parametric filters. In other words, we want to test

whether the local SSFT is able to learn the prescribed parametric spectra directly from the target noise field. Given such heterogeneous target field, the application of the global non-parametric generator will fail to capture the trend in the spectral slope. This can be seen in Fig. 4b, where the noise field has homogeneous characteristics throughout the image and it is confirmed by the black contours representing the resulting local non-parametric 2D autocorrelation functions. Conversely, the local non-parametric generator applied in Fig. 4c closely





reproduces the autocorrelation structure that was prescribed in the target image. The spatial coherence between different windows, i.e. the continuity in absolute values at the edges of consecutive windows, is the result of convolving the white noise field with the amplitude spectrum of the rainfall field only. A reasonable amount of overlapping between windows (e.g. 50%) would be sufficient to obtain a smoother representation of the non-
stationary anisotropy or autocorrelation range in presence of stronger gradients.

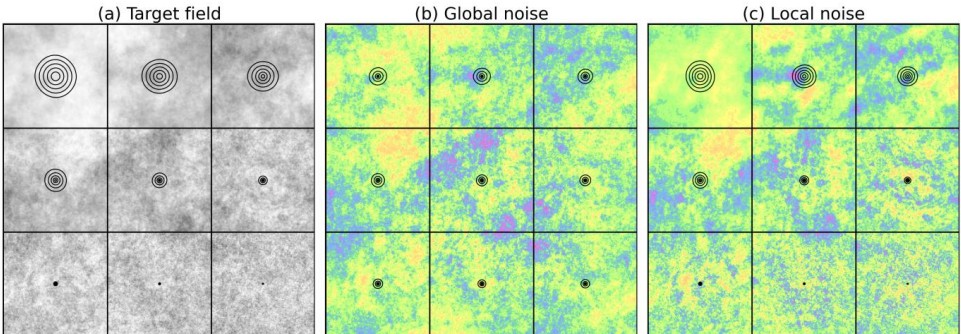

**Figure 4.** (**a**) An example of 512x512 pixel non-stationary correlated Gaussian noise generated using parametric filtering. The spectral slope $\beta$ increases from -3.0 to -1.5 from the top towards the bottom of the image as represented by black contours at 0.5, 0.6, 0.7, 0.8 and 0.9 correlation coefficients. Non-parametric stochastic realisations of the target field using a
global generator (**b**) and a local SSFT generator (**c**) with Hanning windows centred over the grid boxes in the image. All simulations share the same random seed.

As next step, anisotropy was introduced in the simulation of synthetic data. This was done by means of the Generalized Scale Invariance (GSI) model as presented by Niemi *et al.* (2014). The method was firstly introduced by Lovejoy and Schertzer (1985) to describe the scaling behaviour of anisotropy within multifractal
fields. The GSI method has the advantage of being able to model the scale dependence of anisotropy using only few parameters. It should be noted that the parametrisation of anisotropy can also be done with 2D anisotropic autocorrelation functions as can be found in the field of geostatistics. In such cases, the corresponding filter in the Fourier space is obtained with the Wiener-Khintchine theorem (Eq. 2.3).

In a similar way as in Fig. 4, the parameters of the GSI model were spatially varied in order to introduce a
changing anisotropy into a field of correlated Gaussian noise. The result of the simulation is presented in Fig. 5a. We tested again the ability of the global and local non-parametric generators to reproduce the spatial heterogeneities introduced in the target field and the results are presented in Fig. 5b and Fig. 5c. While generally the global generator could only learn and reproduce the same average anisotropy, the local approach managed to localise the distinct patterns of the field. In particular, the method is effective in producing the correct angle of
anisotropy.

In the non-parametric local realisations presented in Fig. 4c and Fig. 5c, on can notice a certain contraction of the contours of the autocorrelation functions compared to the references in the target field. This means that the non-parametric realisations exhibit slightly lower correlations ranges. We believe that this is caused by a combination of minor source of errors associated to the use of the Hanning window and imperfections in the white noise.
Hardly preventable, these minor deviations are the integral part of the method. It should also be noted that some degree of localisation can still be found in the result of the global non-parametric simulations. This is believed to be the effect of the residual phase information contained in the global Fourier spectrum used for the white noise filtering (Sect. 2.2). However, this will not be the case with a parametric filter, as no phase information would be available.





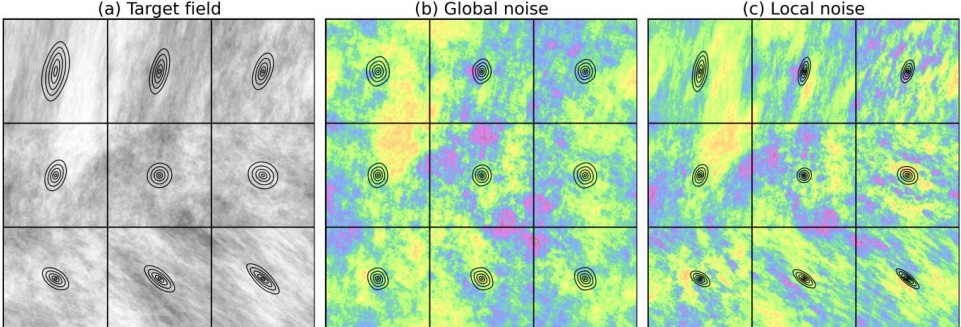

**Figure 5.** (**a**) An example of 512x512 pixel non-stationary correlated Gaussian noise generated using parametric filtering. The GSI parameters (see Niemi *et al.*, 2014 for details) linearly evolve from G(-0.12,-0.12,-0.12) and $I_s$ = 5 at the upper left corner to G(0.12,0.12,0.12) and $I_s$ = 5 at the lower right corner of the image. The black contours show the corresponding autocorrelation functions at 0.5, 0.6, 0.7, 0.8 and 0.9 correlation coefficients. Non-parametric stochastic realisations of the target field using a global generator (**b**) and a local SSFT generator (**c**) with Hanning windows centred over the grid boxes in the image. All simulations share the same random seed.

### 4.1 Effect of window size

The side-length of the window function used to estimate the local Fourier spectrum is expected to affect the accuracy of the estimation. In fact, if a too large window is used, then the localisation gets less and less informative and the assumption of stationarity within the window becomes weaker. The global approach represents a particular case, when the window size equals the image size. On the other hand, if a too small window is used, we lose information at wavelengths larger than the window itself and the filter may become ill-defined due to limited sample size.

In Fig. 6, nonzero synthetic realisations of 512x512 pixels were produced using Eq. 4.1 and β = -3.0, -2.5, -2.0 and -1.5. We then applied the SSFT method with decreasing window sizes of 512, 256, 128 or 64 pixel width to produce 20 stochastic realisations for each of the windows. Finally, the average 1D power spectra of the resulting fields of correlated noise were plotted in Fig. 6a, alongside the benchmark representing the 1D power spectra of the original image. In order to facilitate the comparison, all spectra were centred by removing the mean value. The idea was to investigate the impact of decreasing window sizes on the capacity to reproduce the global power spectra of the original image. The effect of reducing the window size is visible in Fig. 6 as we observe a loss of power for wavelengths larger than the window itself. The magnitude of the deviation is more and more important for smaller window sizes and can therefore represent a limitation in the reproduction of scales larger than the chosen window size. However, Fig. 6 also shows that the effect gets less important as the spectral slope of the original field decreases. This outcome suggests that the choice of the window size, or rather its lower limit, should consider the spatial characteristics of the target field, namely the presence of large scale features. The same is observed with real precipitation fields as discussed later (see Fig. 11).




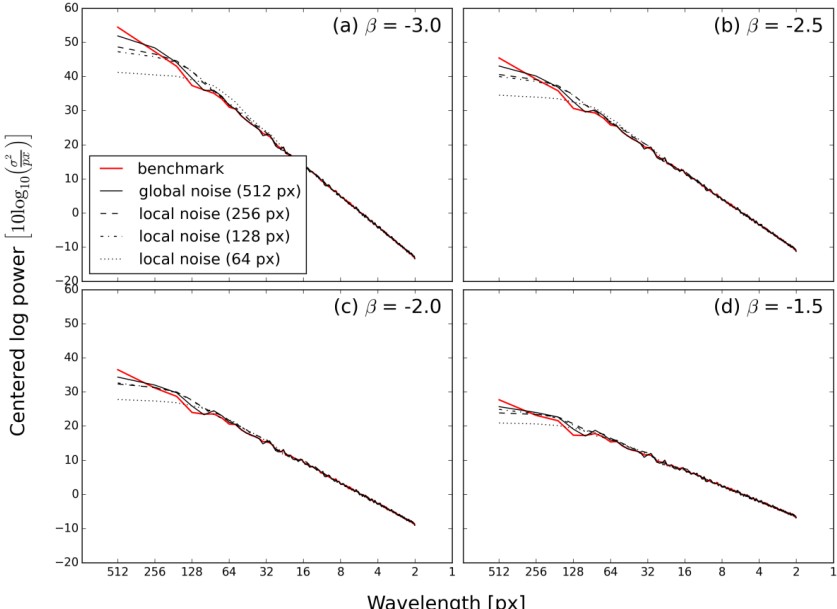

**Figure 6.** The radially averaged 1D centred power spectra of synthetic realisations with spectral slope (**a**) β = -3.0, (**b**) β = -2.5, (**c**) β = -2.0 and (**d**) β = -1.5 (in red). In black are the corresponding average spectra of 20 non-parametric realisations for decreasing window sizes. The same experiment was realized with real precipitation fields (see Fig. 11).

## 5. Weather radar rainfall fields

Precipitation fields from the MeteoSwiss operational radar product for QPE (Germann *et al.*, 2006) were employed for testing the method with real data. In 2015, this weather radar product corresponded to the Cartesian composite of 4 C-band Doppler radars equipped with dual polarization (Germann *et al.*, 2015). It should be noted that the radar on the Weissfluhgipfel became operational only in 2016 and it thus not included in this analysis (Fig. 7).





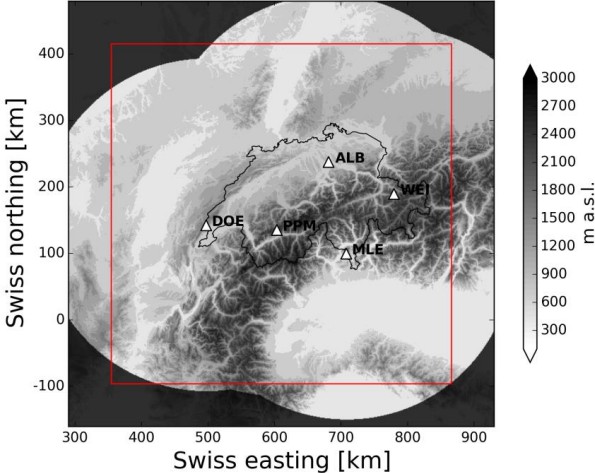

**Figure 7.** The 512x512 km study domain (red box) plotted over the Alpine orography, the Swiss national borders and the radar composite mask. Locations of the 4[th] generation operational MeteoSwiss weather radars are reported as white triangles. ALB: Albis (925 masl), DOE: La Dôle (1675 masl), MLE : Monte Lema (1625 masl), PPM: Pointe de la Plaine Morte (2942 masl), WEI: Weissfluhgipfel (2850 masl, operational since 2016).

The raw volumetric radar data undergo a sophisticated signal processing in order to guarantee the best possible precipitation estimates at ground level in a predominately alpine environment. The processing chain includes ground clutter removal by means of a clutter-elimination decision tree including dual-pol moments, visibility correction, correction for the vertical profile of reflectivity (VPR), global and local bias correction. For more details, the reader is referred to Germann *et al.* (2006).

Spatial resolution of the final Cartesian radar grid is 1 km and the temporal sampling is 5 minutes. In this study, the original images are cropped to a 512x512 km domain as illustrated by the red box in Fig. 7. Rainrates are converted back to reflectivity (dBZ) using the operational MeteoSwiss Z-R relationship:

$$dBZ = 10\log_{10}(316R^{1.5}),$$                    **Eq. 5.1**

where $R$ is the rainrate in mm h$^{-1}$. Given the skewed distribution of rainrates, the logarithmic transformation leads to approximately normally distributed data, which provides better conditions for the computation of the Fourier transform. The rain/no-rain threshold is set at 0.08 mm hr$^{-1}$ (corresponding to 8.54 dBZ). In order to reduce the discontinuity produced by the thresholding, we opted for subtracting 8.54 dBZ to rainy pixels, while keeping all the non-rainy pixels to zero.

The choice of the rainfall threshold and of the value associated to zero rain in dBZ is quite critical and affects the Fourier spectrum, in particular at high spatial frequencies. In fact, a larger step at the rain/no-rain transition has the effect of increasing the power at high frequencies, thus decreasing the absolute value of the spectral slope β. As a consequence, the generated noise fields will have more power at high spatial frequencies (more spatial detail and shorter correlation structure).

### 5.1 Non-stationary rainfall cases

The first case study is taken from an event on 30 March 2015 and is presented in Fig. 8a. The main north-westerly flow is associated with strong orographic blocking on the northern slopes of the Alps, producing persistent stratiform precipitation on a large portion of the country. At the same time, a squall line develops to





the north and rapidly moves to south-east following the displacement of the cold front. These two features display contrasting spatial characteristics: the pre-frontal, orographic precipitation has low intensity and a long range of spatial autocorrelation, while the cold front produces convective activity organized on a straight line with a different orientation angle. The squall line is clearly visible in the 2D power spectrum as a horizontal line

passing through the centre of the spectrum (Fig. 8a, centre). In the 2D autocorrelation function in the bottom panel of Fig. 8a, these two distinct patterns translate at larger scales into a main anisotropy of approximately 45° (counter clockwise angle of rotation from due east) and into a different one at small scales with approximately 0° angle.

The second case study is from 15 May 2015 at 16:00 UTC and is illustrated in Fig. 8b. The south-easterly flow

pushes convective cells to the Southern Alps where they are blocked by the mountain ridge. At the same time, narrow bands of precipitation are distributed on the northern and western portions of the image while moving south-west. Overall, the apparent motion field has a counter clockwise rotation about the centre of the image and is responsible for the appearance of the local anisotropies. These two distinct precipitation structures have pronounced anisotropies at 90° from each other meaning that they cancel out once averaged together over the

field, finally resulting in a fairly isotropic 2D autocorrelation function, as shown in the bottom panel of Fig. 8b.

The third case study belongs to an event on 8 June 2015 at 02:30 UTC (Fig. 9a). The image is characterised by a distinct active line of thunderstorms along the Southern Prealps. This is originated from weak easterly currents bringing humid air towards the Alps that is then destabilised by the transition of an occluded front. Precipitation in the northern side of the Alps is instead associated to a low pressure at ground level. These two main features

have strong differences in both orientation and correlation ranges that are caused by the presence of the Alps.

The fourth and final case study is from 15 June 2015 at 11:45 UTC (Fig. 9b). This case is not characterised by any distinct large scale anisotropy as the precipitation patterns look almost isotropic. Instead, it is easy to notice a north-south gradient in the autocorrelation ranges of the rainfall patterns. In fact, the northern side of the Alps is principally characterised by patches of stratiform precipitation of low intensity, while in the southern part of

the image there are small scales precipitation structures only in correspondence of topographical features.




**Figure 8.** Radar rainfall fields (top row), 2D Fourier spectra zoomed on frequencies > 13 km and rotated by 90° (centre row) and corresponding 2D autocorrelation functions (bottom row) for case studies on (**a**) 30 March 2015 and (**b**) 15 May 2015. The autocorrelation function is obtained as inverse FFT of the 2D power spectrum. The anisotropy is estimated by eigenvalue decomposition of the autocorrelation function using values only inside the dashed region and shifted to start from 0. The eccentricity is obtained as sqrt(1-minor-axis/major-axis) and is comprised between 0 (perfect circle) and 1 (minor-axis << major-axis).





**Figure 9.** Radar rainfall fields (top row), 2D Fourier spectra zoomed on frequencies > 13 km and rotated by 90° (centre row) and corresponding 2D autocorrelation functions (bottom row) for case studies on (**a**) 8 June 2015 and (**b**) 15 June 2015.

## 5.2 Reproduction of local non-stationarities

5 Both global and local noise generators were tested to detect and model the local non-stationarities in the weather radar images presented above. The local SSFT generator was applied with a window side-length of 256 km (i.e. half of the original field size), a Hanning window function and no overlapping. Such settings allow a clear





representation and interpretation of the results. The global FFT generator can be seen as particular case of the local SSFT generator with window side-length equal to the field size, 512 km in this case.

Fig. 10 presents for all case studies the 2D autocorrelation functions within the four windows of 256 km side-length as shown by the grid. The latter is computed and averaged over 20 realisations of global and local noise, while only the first realisation is shown in the background as example. It should be noted that the notion of local is defined by the choice of the window size, as there is the assumption of homogeneity of statistical properties within a single window. The visual comparison shows that for all four cases, only the local approach can effectively reproduce the non-stationarities as defined in the reference images on the left. It is easy to notice how the global generator systematically produces average autocorrelation functions that are driven by the main anisotropies observed in the field. As already discussed in the synthetic examples in Sect. 4, small differences between the reference 2D autocorrelation functions and those computed from the obtained SSFT noise fields are believed to be linked to sources of errors caused by windowed approach, such as the use of the Hanning window. Arguably, even for 256 km window the assumption of stationarity appears as weak and this is noticeable by the presence of abrupt changes between some of the adjacent windows. The use of smaller windows and the introduction of some degree of overlapping between successive windows will guarantee a smoother transition over the field.





**Figure 10.** Localised 2D autocorrelation functions (black contours for 0.5, 0.6, 0.7, 0.8, 0.9 correlation coefficients) for all four case studies (rainrates in black-and-white) and the corresponding non-parametric stochastic simulations using both the global and local generators. The image domain is 512x512 km. All noise fields share the same random seed.





### 5.3 Reproduction of the global power spectrum

The goal of the global FFT-noise generators is to impose the same power spectrum observed in the precipitation radar image to a field of Gaussian white noise. The local SSFT-noise generator has the advantage of being able to localise this operation in order to account for non-stationarities. However, an obvious question is whether the

local SSFT-noise generator is still able to reproduce the global power spectrum, despite representing the precipitation as a convolution of localised Fourier transforms. The analysis over synthetic data (see Fig. 6) showed that there is a loss of power in the spectrum for wavelengths larger than the window size. The same analysis can be done with the precipitation fields presented above, thus introducing all the complexities associated to real data. For this verification we computed the radially averaged 1D power spectra of the target

rainfall field, the global-FFT noise and the local SSFT noise using three different window sizes (264, 128 and 64 km). For the sake of comparison, all the power spectra were centred to have a zero mean.

The 1D power spectra in Fig. 11 show that overall both global and local FFT approaches manage to impose to uncorrelated white noise a similar scaling behaviour as observed in the reference rain analysis. However, it is easy to notice the loss of power at large wavelengths associated with the reduction of the window size.

Moreover, the magnitude of the deviation varies between the four case studies. Interestingly, the last case study in Fig. 11d displays the least significant deviations at large wavelengths, while it is also associated with the lowest spectral slope (i.e. the least negative $\beta$), as the precipitation field is mainly dominated by small scale features (see Fig. 9b). Such case relates well to the synthetic experiment of Sect. 4.1, where we observed lower deviations for increasing (i.e. less negative) $\beta$. Again, these results support the idea that the choice of the optimal

window size is case dependent.

**Figure 11.** In red are represented the 1D centred power spectra of precipitation fields for the four case studies. In black are the corresponding average spectra of 20 non-parametric noise field realisations with decreasing window sizes.





### 5.4 Stochastic rainfall fields by local noise adjustment

A first application of the SSFT generator is related to design storms. In order to dimension the size of hydraulic infrastructure (sewer systems, hydroelectric dams, artificial banks and bridges, etc) it is important to model the maximum expected flood and discharge under different weather situations. The Probable Maximum Precipitation

(PMP) can be computed both by running NWP models using extreme input conditions or by stochastic simulation of a large number of synthetic rainfall fields, e.g. using statistics from extreme events that occurred in the past.

The FFT-noise and SSFT-noise fields already reproduce the power spectrum and spatial autocorrelation function of the rainfall fields, but they do not replicate the same marginal statistics (empirical statistical distribution, mean

and variance, wet area ratio, etc). There are different techniques that are used in order to transform the noise fields so that they reproduce the statistics of rainfall fields. A well-established approach is to threshold the noise field to have the same wet area ratio (WAR) of the target rainfall field and to re-scale the non-zero values to have the same marginal mean precipitation, usually known as "shift and scale" (Pegram and Clothier, 2001a,b; Berenguer *et al.*, 2011). A similar method attempts to match the observed mean and standard deviation of the

precipitation field (Seed *et al.*, 1999, Niemi *et al.*, 2014). More sophisticated methods apply a quantile-quantile transformation to match exactly the same empirical distribution of observed precipitation values, which is also known as probability matching or anamorphosis (Metta *et al.*, 2009; Schleiss *et al.*, 2014).

All these transformation techniques share the same dilemma, namely that changing the marginal statistics of the noise fields perturbs their power spectra (Metta *et al.*, 2009). More precisely, the noise fields accurately

reproduce the power spectrum of the precipitation fields but not their marginal statistics. On the other hand, the simulated precipitation fields accurately reproduce the marginal statistics but the power spectra are slightly perturbed. Following the approach of Pegram and Clothier (2001a,b), a local adjustment of the noise is applied.

1. Observed wet area ratio ($WAR_0$), marginal mean ($MM_0$) and marginal standard deviation ($MSD_0$) are computed based on the observed radar rainfall field, $Z_0$, in dBZ units.

2. From the noise field, $Z_N$, a threshold $T$ is computed as the $(1\text{-}WAR_0)^{\text{th}}$ percentile, such that the number of pixels above or equal to $T$ is the same as the number of wet pixels in the observed field.

3. The marginal mean and marginal standard deviation with respect to $T$ (thus $MM_T$ and $MSD_T$) are computed from $Z_N$.

4. The noise field $Z_N$ and the threshold $T$ are renormalised to match the marginal distribution of $Z_0$ as

follows:

$$Z_N^* = \frac{(Z_N - MM_T)}{MSD_T} MSD_0 + MM_0,$$

$$T^* = \frac{(T - MM_T)}{MSD_T} MSD_0 + MM_0.$$

            **Eq. 5.2**

5. Finally, all values in $Z_N^*$ below $T^*$ are set to zero and the field can be back-transformed into rainrates by inverting the Z-R relationship in Eq. 5.1.

Steps 1 to 5 can be applied at once to the whole image or more locally within smaller windows. In this case, values of $MM_0$, $MSD_0$, $MM_T$, $MSD_T$ and $T^*$ need to be assigned to every pixel by interpolation to avoid the

appearance of discontinuities in the rescaled fields. As final adjustment, a global quantile-quantile transform (probability matching) is applied to reproduce the cumulative distribution of the observed rainrates.





We applied the above procedure at a resolution of 64 km to a number of realisations of correlated Gaussian noise generated using the rainfall field of case study 1 (30 March 2015). The original field and the three stochastic realisations are presented in Fig. 12 for both globally and locally generated noise. The use of a locally modulated noise allows to better reproduce some of the specific features of the original rainfall field, such as the squall line in the upper portion of the image. Not all realisations produced this local feature with the same realism and still some residual noise affects the quality of the output. Nevertheless, we consider the examples in Fig. 12 as a significant improvement in the context of stochastic simulation of precipitation fields.

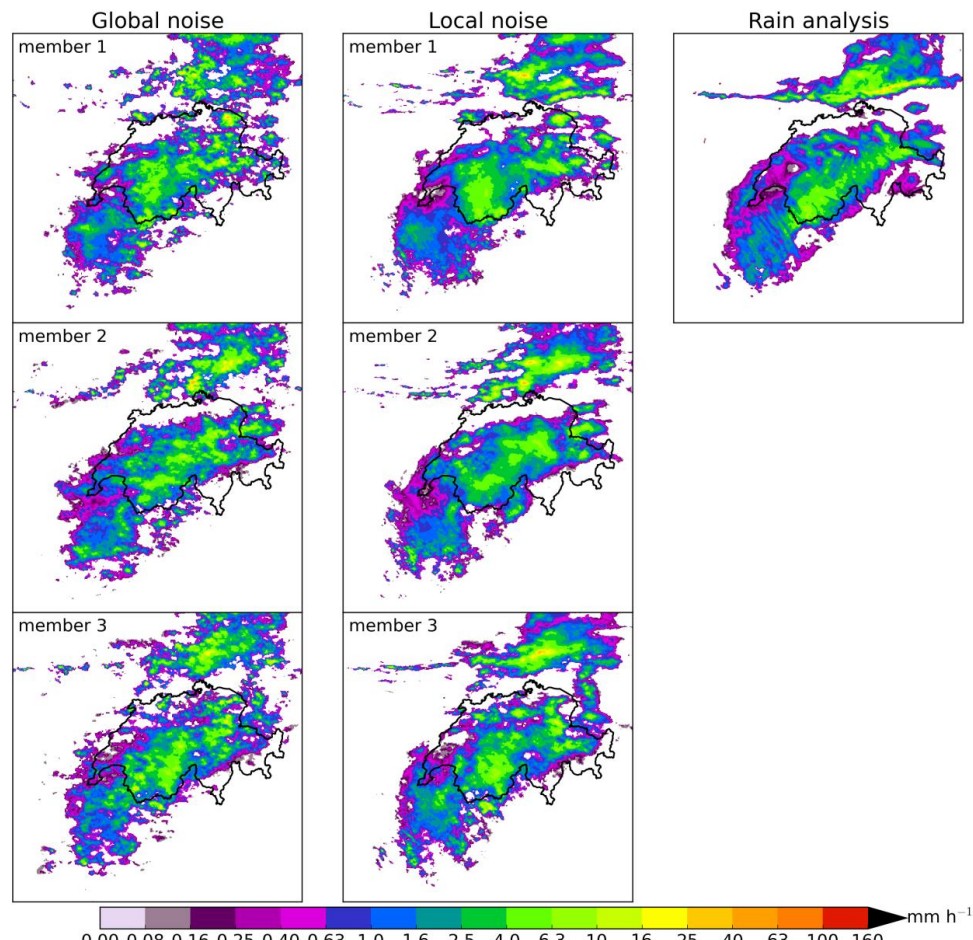

**Figure 12.** Three stochastic realisations conditioned to case study 1 (30 March 2015) after local adjustment at 64 km resolution of correlated global FFT (**left**) and local SSFT (**centre**) Gaussian noise. The top right panel shows the actual observed field. Parameters of the local SSFT generator: window size = 128 km, overlapping = 50%. The national borders of Switzerland are included as spatial reference (black contour).

### 5.5  Imposing the temporal structure of precipitation: the nowcasting example

Probabilistic precipitation nowcasting represents a common application of fields of stochastic noise and it can be helpful to illustrate how to simulate the temporal structure of precipitation. The unknown temporal evolution of





precipitation due to growth and decay processes (often considered in Lagrangian coordinates) is usually modelled by auto-regressive models (AR) of order 1 or 2 (see Pegram and Clothier, 2001a,b; Seed, 2003; Bowler *et al.*, 2006; Berenguer *et al.*, 2011; Seed *et al.*, 2013; Atencia and Zawadzki, 2014). A time series following a first order auto-regressive process (AR(1)) can be constructed as follows:

$$X_{t+1} = \rho X_t + \sqrt{1-\rho^2}\, G(0,1),$$ **Eq. 5.3**

where $t$ is the current time-step, $X$ is the standardised variable to be predicted, $\rho$ is the temporal auto-correlation of the time series and $G(0,1)$ is the shock term, that in the case of a single one-dimensional time series is taken as simple white Gaussian noise with zero mean and unit variance. In the context of nowcasting, $\rho$ is an estimation of the Lagrangian auto-correlation of radar rainfall fields, which is a measure of predictability for the rate of development of rainfall due to growth and decay processes in moving coordinates (Germann *et al.*, 2002). A

cascade of auto-regressive processes can also be considered to account for the scale-dependence of the predictability of precipitation as done in STEPS (Seed, 2003; Bowler *et al.*, 2006).

The problem becomes more complicated when trying to model the temporal evolution of two-dimensional precipitation fields, which requires preserving the joint spatial correlation between different points in the domain. In other words, using a two-dimensional field of white noise as shock term would end up destroying the

spatial structure of the precipitation field after a few time steps. The shock term should serve as a model for the expected forecast errors, which are known to be correlated in both space and time. The classical approach is to filter a field of stationary FFT-noise using an AR(1) model that evolves with the same Lagrangian auto-correlation of the precipitation field. This generates a time series of spatially and temporally correlated noise fields that can be used as shock terms in Eq. 5.3 (Bowler *et al.*, 2006). The auto-regressive filtering of correlated

noise fields allows to condition the future rainfall realisations to the last observed rainfall field and to effectively model the growth of forecast errors by imposing appropriate spatial and temporal correlations.

However, in the presence of non-stationary precipitation fields, it is not adequate to simply use stationary FFT-noise, which does not represent the local characteristics of the field. In fact, as the time progresses the global noise will gradually destroy the local anisotropies and inject too much variability within areas of smoother

stratiform rain, or conversely not enough variability in convective regions. Therefore, we expect the local SSFT-noise to provide a more realistic and locally adaptive shock term.

In order to test this hypothesis we generated two time series of stochastic precipitation fields, one using the global and one using the local shock term. Fig. 13 illustrates an example of stochastic precipitation nowcasting using a simple AR(1) model to simulate the temporal variability of rainfall due to unknown growth and decay

processes. The correlation of the AR(1) process has been set to the Eulerian temporal auto-correlation with the previous radar rainfall field. The visual comparison for increasing lead-times (+5, +30 and +60 minutes) highlights the advantages of using a locally generated shock-term. In fact, the evolution of the image looks more realistic when local noise is employed, as the non-stationarity are preserved both in terms of anisotropy and correlation length. When a globally generated noise is used in the presence of strong non-stationarities, spatial

patterns are quickly destroyed, as the image spatial structure converges towards a global average that looks artificial. In other words, thanks to the local modulation of the shock-term, growth and decay processes take place in a spatially consistent way so that new cells appear with the same anisotropy and spatial autocorrelation as existing nearby cells. On the other hand, a marked improvement in forecast skill is yet to be demonstrated, as performance scores are expected to be driven by the ability of the forecast system to predict the rainfall fraction




and average intensity over a given area. For instance, Gyasi-Agyei (2016) did not find improvements in the kriging interpolation of daily rain gauges when including radar-based locally varying anisotropy.

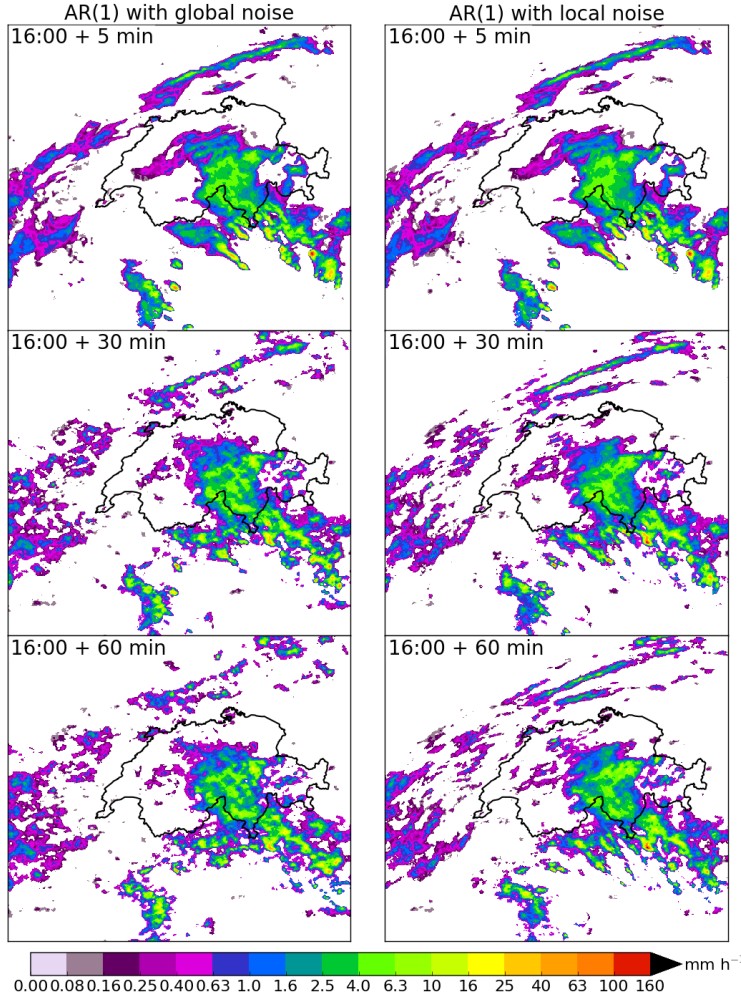

**Figure 13.** Example of simplified stochastic rainfall nowcasting for case study 2 (15 May 2015). Here the advection component is neglected and the movement of the rainfall field is assumed to be zero. Growth and decay processes are simulated stochastically using an AR(1) process with a globally (**left**) or locally (**right**) generated shock term. The lag-1 autocorrelation coefficient was set equal to 0.94. A local adjustment as in Sect. 5.4 was included at 64 km resolution. The national borders of Switzerland are included as spatial reference (black contour). Both simulations share the same random seed.

## 6. Summary and conclusions

In this paper we proposed a novel approach for the generation of non-stationary stochastic rainfall fields. The main idea behind this work consists in the simulation of a stochastic rainfall field in order to preserve not only global spatial correlation properties of the target precipitation field, but also the spatial distribution of its local correlation lengths and anisotropies. This is achieved by replacing the 2D Fourier transform in both non-parametric and parametric conventional filtering of white noise, with the Short-Space Fourier transform.




Namely, instead of convolving the white noise field with the entire precipitation field, we perform the convolution locally, by employing the carefully parametrised moving window. This way, we obtain a spatially correlated noise field, which locally reproduces the spectral properties of the observed rainfall field, i.e. which conserves its local correlation lengths and anisotropies.

We illustrated the potential of the non-stationary generator with synthetic and real precipitation data and demonstrated the advantages of the local approach in presence of non-stationarities. We showed how the non-stationary correlated noise can be locally adjusted to generate realistic stochastic simulations of radar precipitations fields. As a final experiment, we introduced the temporal structure of rainfall in the context of precipitation nowcasting using an auto-regressive process. We demonstrated that the local shock term can help to

preserve the spatial heterogeneities of the original field during the simulation of the temporal evolution of precipitation.

The negative effect of localising the Fourier transform is a loss of power at scales larger than the window size, whose magnitude depends on the global spectral slope. In other words, precipitation fields displaying large scale patterns suffer more from such loss of power. Ways to account for such effect are currently under study. Other

sources of errors associated to the method itself, such as the use of a Hanning window, were found to slightly affect the reproduction of local autocorrelation structure. However, we believe that these limitations are compensated by the benefits introduced by the new approach.

### 7. Future perspectives

The non-stationary stochastic generator was designed in the context of ensemble quantitative precipitation

estimation and nowcasting, blending and downscaling of NWP models, and design storms modelling. As already mentioned in the introduction (Sect. 1.1), several stochastic rainfall models based on the global Fourier filtering of white noise can be reformulated using the local non-stationary SSFT generator.

The residual radar measurement uncertainty can be effectively modelled by the non-stationary generator to obtain QPE ensembles which reproduce the local statistical characteristics and anisotropy of the observed

rainfall fields (Jordan *et al.*, 2003; Ciach *et al.*, 2007; Villarini *et al.*, 2009; Germann *et al.*, 2009). In addition, we believe that current radar rain gauge merging and adjustment techniques (e.g. Sideris *et al.*, 2014) can be further extended to generate QPE ensembles conditioned onto the rain gauges, for example by following the conditional merging approach of Sinclair and Pegram (2005).

A demonstrative application of the local SSFT generator for stochastic precipitation nowcasting was presented in

Sect. 5.5. A complete nowcasting model would also need an optical flow technique for the Lagrangian extrapolation of the stochastic radar rainfall fields. In the example presented in this paper the stochastic perturbations are generated by using the same set of local Fourier spectra estimated at analysis time for all forecast lead times. An interesting scientific question would be to analyse the persistence and predictability of the local Fourier spectra, which in turn will control the future evolution of the local properties of the stochastic

rainfall fields. This question is particularly important to design ensemble precipitation nowcasting systems that better represent the forecast uncertainty.

For longer forecast ranges (e.g. 24-48 hours) the statistical properties of the radar rainfall field at analysis time become even more obsolete and should not be used to generate the stochastic perturbations. Current FFT-based downscaling techniques simplify the problem by using a fixed climatological isotropic parametric filter (Rebora





*et al.*, 2006a; Pierce *et al.*, 2010). As a consequence all the downscaled NWP fields share the same small scale precipitation features, independently on the NWP field that needs to be downscaled. A more flexible approach would involve the generation of the stochastic perturbations that adapt to the local properties and anisotropy of the NWP forecast.

Finally, the extension of the non-stationary generator for design storms would need additional components to model the storm arrival process (duration of dry and wet spells) and the temporal evolution of the mean areal statistics (global IMF and WAR), see e.g. Paschalis *et al.* (2013).

The SSFT generator makes a certain number of assumptions and can be further developed. In its current implementation the rainfall field is convolved by rectangular or Hanning windows of fixed size, which assumes

stationarity of rainfall properties within the window. However, the dimension or length of stationary regions within a rainfall field could also depend on spatial location. For example, there may be a small region of 50x50 km$^2$ characterised by cellular convection in one part of the domain and another region of 200x100 km$^2$ with frontal stratiform rain. Adaptive windows that dilate and contract according to the length of the stationary region could be an interesting extension of the method. Given the link between the Short-Time Fourier Transform and

the wavelet decomposition (Kovačević *et al.*, 2013), it would be interesting to develop a non-stationary wavelet-based stochastic noise generator for comparison with the local SSFT approach.

A practical problem of the local Fourier analysis is the inevitable loss of power at wavelengths larger than the window size, which becomes more accentuated for decreasing window sizes. For smaller window sizes, the local Fourier spectrum could be parametrised and the scaling behaviour extrapolated to scales larger than the window

size to compensate for the loss of power.

Finally, the Fourier filtering of white Gaussian noise is a particular case of a broader class of continuous-in-scale multifractal simulation methods (Lovejoy and Schertzer, 2010). Instead of filtering white Gaussian noise, the universal multifractal simulation techniques are based on the filtering of Lévy noise, which provides more flexibility for modelling of multiscaling precipitation fields. An interesting extension of the method would be to

exploit the SSFT approach for non-stationary continuous-in-scale multifractal simulation.



**Acknowledgments**

This study was supported by the Swiss National Science Foundation Ambizione project "Precipitation attractor from radar and satellite data archives and implications for seamless very short-term forecasting" (PZ00P2_161316). We thank Prof. Isztar Zawadzki for the fruitful discussions.



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
