# Peer review of "A non-stationary stochastic ensemble generator for radar rainfall fields based on the Short-Space Fourier Transform"

_Hydrology and Earth System Sciences, 2016_

## Referee Comment (RC1) · G. Pegram (Referee) · 2 Feb 2017

What a pleasure to read an innovative, well written, well-argued and well-constructed paper on the difficult procedure of generating and forecasting radar rainfall fields which exhibit spatial and temporal inhomogeneity. It is so rich in ideas it could have been written as a series, but the authors are at pains to keep the reader properly informed at a moderate pace with full explanations of their thinking and how the pieces fit together.

Section 1.1 is a very fair review of the current status of radar-rainfall forecasting, summarised in the first sentence of section 1.2, which is: "A major limitation and concern of all the cited stochastic generators is that they assume spatial stationarity, i.e. uniformity of the generator across space." and time. This sets the stage and got my attention.

[Figure]

I want to highlight some of the passages that made me sit up, with the remarks I made in the margin as I was reading through, located by page and line number in the pdf.

p8:21 "...we mostly alter the spectral phase of a rainfall field, without significantly modifying the amplitude". That's neat. p16:19-23 "The choice of the rainfall threshold and of the value associated to zero rain in dBZ is quite critical and affects the Fourier spectrum, in particular at high spatial frequencies. In fact, a larger step at the rain/no-rain transition has the effect of increasing the power at high frequencies, thus decreasing the absolute value of the spectral slope 'beta'. As a consequence, the generated noise fields will have more power at high spatial frequencies (more spatial detail and shorter correlation structure)." I'd never thought of that - what a good point and solution. P27:33 "An interesting scientific question would be to analyse the persistence and predictability of the local Fourier spectra, which in turn will control the future evolution of the local properties of the stochastic rainfall fields. This question is particularly important to design ensemble precipitation nowcasting systems that better represent the forecast uncertainty". That's a very important and wise observation.

Finally, Section 7 provides a good summary and the last paragraph points the way forward.

This is one of the best papers I have reviewed for a very long time and I unreservedly recommend its publication in HESS.

Geoff Pegram 2 February 2017
* * *
For the authors' convenience I am noting some small grammatical changes that should be made to the text.

P9:26 remove 'an' P11:15 change 'cantering' to 'centring' P13:26 change 'on' to 'one' P16:17 change 'to' to 'in' P18 In axes of spectra, change 'wavelenght' to 'wavelength' P18:6 change 'comprised' to 'constrained' P22:18 change 'Such' to 'This' P23:1

change 'by' to 'using' P25:33 change 'are' to 'is'

---

## Author Comment (AC1) · 13 Feb 2017

We would like to thank Prof. Pegram for the very encouraging comments to our manuscript and his corrections, which we readily included.

We hope that the conceptual simplicity of our contribution and the fact that it builds on an already widely established approach will help to arouse interest among the precipitation nowcasting community and foster new research in non-stationary stochastic generators.

With respect to the comment on the effect of the rain/no-rain transition on the slope of the Fourier spectrum, we have investigated the sensitivity of the spectral slope to

the magnitude of the rain/no-rain transition. One such analysis is presented in Fig. 1, where four different settings were compared: with/without a step (i.e. 'Zeros = 0.0 dBZ' or 'Zeros = 8.5 dBZ') and with/without Gaussian smoothing ($\sigma$=2) of the edges (i.e. 'Filter = 2' or 'Filter = No'). The location of the scaling break was also optimized by maximizing the correlation coefficients of the two linear fits. It can be noticed how the spectral slope $\beta$1, which relates to the larger scales, is almost insensitive to the rain/no-rain transition. Instead, the spectral slope $\beta$2 largely depends on the magnitude of such transition. Such outcome was somewhat expected, as the discontinuity in the signal causes the rise in the energy of the higher frequency components (smaller scales) of the Fourier spectrum. In fact, if we ignore the effects of an additional filtering of the discontinuity (i.e. filter = No), it can be seen that $\beta$2 = -3.70 when the discontinuity is largest. Instead, $\beta$2 = -4.25 if the discontinuity is reduced by assigning the value of the threshold to all zero rain pixels. On this basis, we concluded that it is important to reduce the discontinuity of the rain/no-rain transition in order to limit an injection of variability at small scales which might be in fact artificial, as already highlighted in Bowler et al. (2006). This result is in line with what already included in our manuscript, but if you consider that a more in depth analysis like the one presented above is needed, we could certainly try to integrate Fig. 1 in the paper.

REVIEWER 1:

P9:26 remove 'an' P11:15 change 'cantering' to 'centring' P13:26 change 'on' to 'one' P16:17 change 'to' to 'in' P18 In axes of spectra, change 'wavelenght' to 'wavelength' P18:6 change 'comprised' to 'constrained' P22:18 change 'Such' to 'This' P23:1 change 'by' to 'using' P25:33 change 'are' to 'is'

ANSWER:

We included all the above corrections in the manuscript.

REFERENCES

Bowler, N. E. H., Pierce, C. E., and Seed, A. W.: STEPS: A probabilistic precipitation forecasting scheme which merges an extrapolation nowcast with downscaled NWP, Q. J. Roy. Meteor. Soc., 132, 2127–2155, 2006.
* * *
**1D power spectrum for 2015-03-30 05:20:00**

WAR = 34.9 %
MM = 1.906 mm/hr

Reflectivity field power $\left[10\log_{10}\left(\frac{dBZ^2}{km}\right)\right]$

Wavelenght [km]

| | |
|---|---|
| —— | Zeros = 0.0 dBZ, filter = No. $\beta_1$ = -2.63, $\beta_2$ = -3.70, break at 11 km |
| — · | Zeros = 0.0 dBZ, filter = 2 . $\beta_1$ = -2.51, $\beta_2$ = -4.84, break at 22 km |
| —— | Zeros = 8.5 dBZ, filter = No. $\beta_1$ = -2.69, $\beta_2$ = -4.25, break at 10 km |
| — · | Zeros = 8.5 dBZ, filter = 2 . $\beta_1$ = -2.48, $\beta_2$ = -4.56, break at 22 km |

**Fig. 1.** Analysis of the sensitivity of spectral slopes $\beta_1$, $\beta_2$ and scale break to the strength of the rain/no-rain discontinuity.

---

## Referee Comment (RC2) · Anonymous Referee #2 · 6 Mar 2017

GENERAL COMMENTS AND RECOMMENDATION

The manuscript illustrates the use of the Short-Space Fourier Transform to simulate non-stationary rainfall fields. This technique has been applied to Switzerland to simulate the properties of four radar rainfall fields.

This is a very interesting topic, and the proposed approach might be able to solve some of the limitations of the currently existing methods for characterizing the uncertainties in radar-based QPE and nowcasting (among other topics) using the concepts of stochastic simulation.

The manuscript is well written and organized and provides insightful discussion and,

consequently, I recommend the publication of the manuscript in Hydrology and Earth System Sciences. The authors could also take into account the following minor comments.

MINOR COMMENTS

1) I find the presentation of the technique somewhat unbalanced: while most of the concepts presented in detail in section 2 can be found in a number of books and are, in general, well-known (it could be moved to an Appendix), almost no detail is provided about the method used to impose the local anisotropy (based on the Generalized Scale Invariance model; Niemi et al., 2014). I would strongly suggest to add a brief description (if necessary, in an Appendix, as well).

2) The figures should be sequentially cited in the text. Currently, Figs. 8 and 9 are cited in page 7 (before first citation of Fig. 2), and Fig. 11 is cited in page 14 before Fig. 7.

3) Figs. 8 and 9. It is unclear to me why the authors have chosen to rotate the 3D power spectra by 90° and use a decreasing y axis to display the 2-D autocorrelation functions. Why is it better to use these configurations?

4) I miss the color palette in Figs. 3 – 5 and 10. It is clear that the simulated fields have arbitrary units, but this could be explicitly stated in the text.

SPECIFIC COMMENTS

1) Page 1, line 11. "Differences" could be replaced by "variability".

2) Page 4, line 17. To my knowledge, Ciach et al. (2007) did not propose the use of any stochastic noise generator. The sentence "A major limitation and concern of all the cited stochastic generators is that they assume spatial stationarity..." (page 4, lines 26-30) might be misleading because some of the references provided in section 1.1 (e.g. Germann et al., 2009; Villarini et al. 2009) did not assume spatial stationarity of the rainfall field.

3) Page 6, lines 27 – 34 and elsewhere. The term "spectrum" is used indistinctively to refer to the Fourier spectrum, X(f), and to the power spectral density, S(f). For clarity, it could be better to use it for S(f).

4) Page 9, lines 26-30. At first, I found this paragraph a little misleading: although the title of the section is "Short-space Fourier transform", this first paragraph (and up to Page 10, line 4) focuses on the time-frequency signal analysis.

---

## Short Comment (SC1) · Dear Daniele et al · 14 Mar 2017

Dear Daniele et al

It is quite good idea what you are proposing in the paper to locally filter rainfall fields and effectively dealing with their natural spatial anisotropy.

I would like to comment only on the core of your proposal that it is the use of a weighting window in the space domain. I am wondering if the reason of the contraction/differences in the local autocorrelation functions reported in your paper may be caused by the particular type of filtering window chosen to illustrate your technique. The standard Hanning window used in the paper may smooth too much the observed

rainfall fields in particular at the edges of the window and therefore may not capture enough of the structure of the rainfall.

As alternative to better capture the rainfall structures across the whole window, the Hanning window may have a flat top until a given distance from the center of the window and then allow the decay of the filter as per your original Hanning window (See Figure 1).

In this way you will have a significant area of the window preserving rainfall values as they are in the original rainfall field (weighted with a Factor=1) and then weighting values moving down from this "flat plateau" area towards the edges of the window with similar decay than your original (standard) Hanning window (See Figure 2).

The 2D "top flat" Hanning window presented in Figure 2 was made by rotation of the 1D "flat top" Hanning window (Figure 1) but a outer product may be applied as well in the same way that it was used for the "standard" Hanning window in the paper to extend ever further towards the edges the non-zero areas.

I believe that if you make comparisons between the auto-correlation functions calculated using a "standard" Hanning window and a "flat top" Hanning window, the later will produce "noisier" contours and less contracted autocorrelations than the former, probably because a better representation of the rainfall structure within the window.

The selection of which is the best weighting window to filter rainfall fields by applying the SSFT technique is so interesting and definitely worth a separate paper. The best filtering window will probably depend of the characteristics of the particular problem to solve using SSFT and also their properties in the frequency domain.

This is just a comment on how the innovative method proposed in this paper may be improved and therefore it does not invalidate the rest of the results presented here.

Well done Daniele and team.

Kind Regards

Carlos Velasco-Forero

[Figure]

[Figure]

[Figure]

**Fig. 1.**

[Figure]

[Figure]

**Fig. 2.**

---

## Short Comment (SC2) · 20 Mar 2017

Dear Carlos

Many thanks for your detailed comment on the use of the Hanning window for the localisation of the 2D signal in space. Your concern that the shape of the window might alter the reproduction of the correct power spectrum does make a lot of sense. During our analysis, we tested the method with some of the most common window functions (namely the rectangular, Bartlett, Hanning, Hamming, Blackman and Kaiser windows) and found the Hanning to deliver the best results (most of the others were easily excluded since they produced strong discontinuities at the edges).

Your idea to use a top flat window to capture more of the rainfall field and hence improve on the above is really relevant. As you suggest, a dedicated study on the spectral properties of different window types and their impacts on the reproduction of the local spatial correlation structure of the rainfall field might bring some insights on choice of the optimal window function and help to reduce some of the artefacts that are currently associated with the method.

Following your Short Comment, we took the opportunity to add a small paragraph in the future perspective section in our manuscript. We now mention your idea of using a top flat window and more generally the relevance of investigating in detail the choice of the window type.

Kind regards,

Daniele Nerini

---

## Author Comment (AC2) · 20 Mar 2017

**Response to the interactive comment from Reviewer #2, an anonymous reviewer**

Daniele Nerini et al.
daniele.nerini@meteoswiss.ch

**MINOR COMMENTS**

*1) I find the presentation of the technique somewhat unbalanced: while most of the concepts presented in detail in section 2 can be found in a number of books and are, in general, well-known (it could be moved to an Appendix), almost no detail is provided about the method used to impose the local anisotropy (based on the Generalized Scale Invariance model; Niemi et al., 2014). I would strongly suggest to add a brief description (if necessary, in an Appendix, as well).*

Thank you for raising this point. Our intention was to provide the reader with the necessary basic definitions that are relevant to understand the transition from a classical approach to the Short-Space Fourier transform. By subdividing the theoretical sections into clear subjects, we give the reader the option to skip those parts he or she is already familiar with.

It is important to mention that we only used a non-parametric stochastic generator, i.e. we filter the white noise field using the actual local Fourier transform of the rainfall field. The fitting of a GSI model as in Niemi et al. (2014) is not employed in our study. We only implemented a GSI model and defined arbitrary parameters to test the SSFT approach with synthetic data. For these reasons, we decided not to dedicate a section for a detailed explanation of the GSI model, but we tried to better specify how we employed it.

> *Page 13:* **As next step, the method was tested in its capacity to reproduce the locally varying anisotropy of a synthetic target image. This target image was  produced by means of the Generalized Scale Invariance (GSI) model as presented by Niemi et al. (2014). […].**
> **In a similar way as in Fig. 4, the set of arbitrary parameters of the GSI model  was spatially varied in order to  produce a target image with changing anisotropy .**

*2) The figures should be sequentially cited in the text. Currently, Figs. 8 and 9 are cited in page 7 (before first citation of Fig. 2), and Fig. 11 is cited in page 14 before Fig. 7.*

For the sake of consistency, we removed the whole reference to Figs. 8 and 9 in page 7 and replaced the reference to Fig.11 in page 14 with a single reference to Section 5.3.

*3) Figs. 8 and 9. It is unclear to me why the authors have chosen to rotate the 3D power spectra by 90 degrees and use a decreasing y axis to display the 2-D autocorrelation functions. Why is it better to use these configurations?*

The rotation of the 2D power spectra by 90 degrees in Figs. 8 and 9 is motivated by the wish to improve the interpretability of the anisotropy observed in the power spectra. In fact, the rotation allows a direct comparison to the original radar fields and to the spatial autocorrelation function. Thus, the idea is to help the reader to more easily connect the structures observable in the geographical space with the representation in the Fourier space. We modified the figures' caption in order to motivate our choice:

*Caption of Fig. 8:* **Radar rainfall fields (top row), 2D Fourier spectra zoomed on wavelengths > 13 km and rotated by 90° (centre row) and corresponding 2D autocorrelation functions (bottom row)  […]. The 90° rotation is performed in order to align the anisotropies of the 2D spectra and spatial autocorrelation functions.**

Conversely, the inverted y-axis in the 2D autocorrelation function appears to be a simple mistake in our codes: the orientation of the figures is correct, but the axis labels were inverted. We corrected the error in all concerned figures.

*4) I miss the color palette in Figs. 3 – 5 and 10. It is clear that the simulated fields have arbitrary units, but this could be explicitly stated in the text.*

Yes, these are arbitrary units. Specifically, the values belong to the standard normal distribution. As suggested, we both included colorbars in the concerned figures and a short explanation in the captions:

> *Captions of Figs. 3-5 and 10:* **All noise fields have been drawn from the standard normal distribution and share the same random seed.**

**SPECIFIC COMMENTS**

*1) Page 1, line 11. "Differences" could be replaced by "variability".*

We changed the term as suggested.

*2) Page 4, line 17. To my knowledge, Ciach et al. (2007) did not propose the use of any stochastic noise generator. The sentence "A major limitation and concern of all the cited stochastic generators is that they assume spatial stationarity: : :" (page 4, lines 26-30) might be misleading because some of the references provided in section 1.1 (e.g. Germann et al., 2009; Villarini et al. 2009) did not assume spatial stationarity of the rainfall field.*

Thanks for this remark. We moved the citation of Ciach et al. (2007) to the previous sentence where we mention the residual radar measurement uncertainty.
We also agree that our original statement was somehow misleading, considering that not all cited references include the assumption of spatial stationarity, as correctly pointed out by the reviewer. We changed it as follows:

> *Page 4:* ** Apart from few exceptions, the stochastic generators presented above assume spatial stationarity, i.e. uniformity of the generator across space.**

*3) Page 6, lines 27 – 34 and elsewhere. The term "spectrum" is used indistinctively to refer to the Fourier spectrum, X(f), and to the power spectral density, S(f). For clarity, it could be better to use it for S(f).*

Following the reviewer's suggestion, we kept using the term power spectrum as a synonym for power spectral density. The complex Fourier amplitude-phase spectrum is now referred to as complex Fourier representation.

*4) Page 9, lines 26-30. At first, I found this paragraph a little misleading: although the title of the section is "Short-space Fourier transform", this first paragraph (and up to Page 10, line 4) focuses on the time-frequency signal analysis.*

We included an introductory sentence to Section 3.1 in order to limit any possible confusion.

> *Page 9:* **The concept of Short-Space Fourier Transform is introduced through its more intuitive 1D temporal equivalent and then extended to the 2D spatial case.**

**References**

Ciach, G. J., Krajewski, W. F., and Villarini, G.: Product-error-driven uncertainty model for probabilistic quantitative precipitation estimation with NEXRAD data, J. Hydrometeorol., 8(6), 1325–1347, 2007.

Niemi, T. J., Kokkonen, T., and Seed, A. W.: A simple and effective method for quantifying spatial anisotropy of time series of precipitation fields, Water Resour. Res., 50, 5906–5925, doi:10.1002/2013WR015190, 2014.

Germann, U., Berenguer, M., Sempere-Torres, D., and Zappa, M.: REAL—Ensemble radar precipitation estimation for hydrology in a mountainous region, Q. J. Roy. Meteor. Soc., 135, 445–456, 2009.

Villarini, G., Krajewski, W. F., Ciach, G. J., and Zimmerman, D. L.: Product-error-driven generator of probable rainfall conditioned on WSR-88D precipitation estimates, Water Resour. Res., 45, W01404, doi:10.1029/2008WR006946, 2009.

---

## Author Response (AR1)

HYDROLOGY AND EARTH SYSTEM SCIENCES

**A non-stationary stochastic ensemble generator for radar rainfall fields based on the Short-Space Fourier Transform**

D. Nerini, N. Besic, I. Sideris, U. Germann and L. Foresti

Dear reviewers, Dear Editor,

We would like to thank the reviewers for their help in improving the manuscript. We did our best to address all their suggestions in the latest version of the manuscript.

The questions (**m** - minor and **S** - specific) of reviewers are reported in italic font. Quoted modifications of the manuscript are highlighted in bold font.

**1  Referee report #1**

*What a pleasure to read an innovative, well written, well-argued and well-constructed paper on the difficult procedure of generating and forecasting radar rainfall fields which exhibit spatial and temporal inhomogeneity. It is so rich in ideas it could have been written as a series, but the authors are at pains to keep the reader properly informed at a moderate pace with full explanations of their thinking and how the pieces fit together.*

*Section 1.1 is a very fair review of the current status of radar-rainfall forecasting, summarised in the first sentence of section 1.2, which is: "A major limitation and concern of all the cited stochastic generators is that they assume spatial stationarity, i.e. uniformity of the generator across space." and time. This sets the stage and got my attention.*

*I want to highlight some of the passages that made me sit up, with the remarks I made in the margin as I was reading through, located by page and line number in the pdf. p8:21 "...we mostly alter the spectral phase of a rainfall field, without significantly modifying the amplitude". That's neat. p16:19-23 "The choice of the rainfall threshold and of the value associated to zero rain in dBZ is quite critical and affects the Fourier spectrum, in particular at high spatial frequencies. In fact, a larger step at the rain/no-rain transition has the effect of increasing the power at high frequencies, thus decreasing the absolute value of the spectral slope 'beta'. As a consequence, the generated noise fields will have more power at high spatial frequencies (more spatial detail and shorter correlation structure)." I'd never thought of that - what a good point and solution. P27:33 "An interesting scientific question would be to analyse the persistence and predictability of the local Fourier spectra, which in turn will control the future evolution of the local properties of the stochastic rainfall fields. This question is particularly important to design ensemble precipitation nowcasting systems that better represent the forecast uncertainty". That's a very important and wise observation.*

*Finally, Section 7 provides a good summary and the last paragraph points the way forward.*

*This is one of the best papers I have reviewed for a very long time and I unreservedly recommend its publication in HESS.*

*Geoff Pegram 2 February 2017*

**m1:** *P9:26 remove 'an' P11:15 change 'cantering' to 'centring' P13:26 change 'on' to 'one' P16:17 change 'to' to 'in' P18 In axes of spectra, change 'wavelenght' to 'wavelength' P18:6 change 'comprised' to 'constrained' P22:18 change 'Such' to 'This' P23:1 change 'by' to 'using' P25:33 change 'are' to 'is'*

Acknowledged and done.

**2  Referee report #2**

*The manuscript illustrates the use of the Short-Space Fourier Transform to simulate non-stationary rainfall fields. This technique has been applied to Switzerland to simulate the properties of four radar rainfall fields.*

*This is a very interesting topic, and the proposed approach might be able to solve some of the limitations of the currently existing methods for characterizing the uncertainties in radar-based QPE and nowcasting (among other topics) using the concepts of stochastic simulation.*

*The manuscript is well written and organized and provides insightful discussion and, consequently, I recommend the publication of the manuscript in Hydrology and Earth System Sciences. The authors could also take into account the following minor comments.*

**m1:** *I find the presentation of the technique somewhat unbalanced: while most of the concepts presented in detail in section 2 can be found in a number of books and are, in general, well-known (it could be moved to an Appendix), almost no detail is provided about the method used to impose the local anisotropy (based on the Generalized Scale Invariance model; Niemi et al., 2014). I would strongly suggest to add a brief description (if necessary, in an Appendix, as well).*

Thank you for raising this point. Our intention was to provide the reader with the necessary basic definitions that are relevant to understand the transition from a classical approach to the Short-Space Fourier transform. By subdividing the theoretical sections into clear subjects, we give the reader the option to skip those parts he or she is already familiar with.

It is important to mention that we only used a non-parametric stochastic generator, i.e. we filter the white noise field using the actual local Fourier transform of the rainfall field. The fitting of a GSI model as in Niemi et al. (2014) is not employed in our study. We only implemented a GSI model and defined arbitrary parameters to test the SSFT approach with synthetic data. For these reasons, we decided not to dedicate a section for a detailed explanation of the GSI model, but we tried to better specify how we employed it.

**Page 14: As next step, the method was tested in its capacity to reproduce the locally varying anisotropy of a synthetic target image. This target image was produced by means of the Generalized Scale Invariance (GSI) model as presented by Niemi et al. (2014). [...]. In a similar way as in Fig. 4, the set of arbitrary parameters of the GSI model was spatially varied in order to produce a target image with changing anisotropy.**

**m2:** *The figures should be sequentially cited in the text. Currently, Figs. 8 and 9 are cited in page 7 (before first citation of Fig. 2), and Fig. 11 is cited in page 14 before Fig. 7.*

For the sake of consistency, we removed the whole reference to Figs. 8 and 9 in page 7 and replaced the reference to Fig.11 in page 14 with a single reference to Section 5.3.

**m3:** *Figs. 8 and 9. It is unclear to me why the authors have chosen to rotate the 3D power spectra by 90 degrees and use a decreasing y axis to display the 2-D autocorrelation functions. Why is it better to use these configurations?*

The rotation of the 2D power spectra by 90 degrees in Figs. 8 and 9 is motivated by the wish to improve the interpretability of the anisotropy observed in the power spectra. In fact, the rotation allows a direct comparison to the original radar fields and to the spatial autocorrelation function. Thus, the idea is to help the reader to more easily connect the structures observable in the geographical space with the representation in the Fourier space. We modified the figures' caption in order to motivate our choice.

**Caption of Fig. 8: Radar rainfall fields (top row), 2D Fourier spectra zoomed on wavelengths $> 13$ km and rotated by $90°$ (centre row) and corresponding 2D autocorrelation functions (bottom row) for case studies on (a) 30 March 2015 and (b) 15 May 2015. The autocorrelation function is obtained as inverse FFT of the 2D power spectrum. The $90°$ rotation is performed in order to align the anisotropies of the 2D spectra and spatial autocorrelation functions. The anisotropy is estimated by eigenvalue decomposition of the autocorrelation function using values only inside the dashed region and shifted to start from 0.**

Conversely, the inverted y-axis in the 2D autocorrelation function appears to be a simple mistake in our codes: the orientation of the figures is correct, but the axis labels were inverted. We corrected the error in all concerned figures.

**m4:** *I miss the color palette in Figs. 3 - 5 and 10. It is clear that the simulated fields have arbitrary units, but this could be explicitly stated in the text.*

Yes, these are arbitrary units. Specifically, the values belong to the standard normal distribution. As suggested, we both included colorbars in the concerned figures and a short explanation in the captions.

**Captions of Figs. 3-5 and 10:** **All noise fields have been drawn from the standard normal distribution and share the same random seed.**

**S1:** *Page 1, line 11. "Differences" could be replaced by "variability".*

We changed the term as suggested.

**S2:** *Page 4, line 17. To my knowledge, Ciach et al. (2007) did not propose the use of any stochastic noise generator. The sentence "A major limitation and concern of all the cited stochastic generators is that they assume spatial stationarity" (page 4, lines 26-30) might be misleading because some of the references provided in section 1.1 (e.g. Germann et al., 2009; Villarini et al. 2009) did not assume spatial stationarity of the rainfall field.*

Thanks for this remark. We moved the citation of Ciach et al. (2007) to the previous sentence where we mention the residual radar measurement uncertainty. We also agree that our original statement was somehow misleading, considering that not all cited references include the assumption of spatial stationarity, as correctly pointed out by the reviewer. We changed it as follows:

**Page 4:** **Apart from few exceptions, the stochastic generators presented above assume spatial stationarity, i.e. uniformity of the generator across space.**

**S3:** *Page 6, lines 27-34 and elsewhere. The term "spectrum" is used indistinctively to refer to the Fourier spectrum, $X(f)$, and to the power spectral density, $S(f)$. For clarity, it could be better to use it for $S(f)$.*

Following the reviewer's suggestion, we kept using the term power spectrum as a synonym for power spectral density. The complex Fourier amplitude-phase spectrum is now referred to as complex Fourier representation.

**S4:** *Page 9, lines 26-30. At first, I found this paragraph a little misleading: although the title of the section is "Short-space Fourier transform", this first paragraph (and up to Page 10, line 4) focuses on the time-frequency signal analysis.*

We included an introductory sentence to Section 3.1 in order to limit any possible confusion.

**Page 9:** **The concept of Short-Space Fourier Transform is introduced through its more intuitive 1D temporal equivalent and then extended to the 2D spatial case.**

**3   Short Comment #1**

*Dear Daniele et al*

*It is quite good idea what you are proposing in the paper to locally filter rainfall fields and effectively dealing with their natural spatial anisotropy.*

*I would like to comment only on the core of your proposal that it is the use of a weighting window in the space domain. I am wondering if the reason of the contraction/ differences in the local autocorrelation functions reported in your paper may be caused by the particular type of filtering window chosen to illustrate your technique. The standard Hanning window used in the paper may smooth too much the observed rainfall*

*fields in particular at the edges of the window and therefore may not capture enough of the structure of the rainfall.*

*As alternative to better capture the rainfall structures across the whole window, the Hanning window may have a flat top until a given distance from the center of the window and then allow the decay of the filter as per your original Hanning window (See Figure 1).*

*In this way you will have a significant area of the window preserving rainfall values as they are in the original rainfall field (weighted with a Factor=1) and then weighting values moving down from this "flat plateau" area towards the edges of the window with similar decay than your original (standard) Hanning window (See Figure 2).*

*The 2D "top flat" Hanning window presented in Figure 2 was made by rotation of the 1D "flat top" Hanning window (Figure 1) but a outer product may be applied as well in the same way that it was used for the "standard" Hanning window in the paper to extend ever further towards the edges the non-zero areas.*

*I believe that if you make comparisons between the auto-correlation functions calculated using a "standard" Hanning window and a "flat top" Hanning window, the later will produce "noisier" contours and less contracted autocorrelations than the former, probably because a better representation of the rainfall structure within the window. The selection of which is the best weighting window to filter rainfall fields by applying the SSFT technique is so interesting and definitely worth a separate paper. The best filtering window will probably depend of the characteristics of the particular problem to solve using SSFT and also their properties in the frequency domain.*

*This is just a comment on how the innovative method proposed in this paper may be improved and therefore it does not invalidate the rest of the results presented here.*

*Well done Daniele and team.*

*Kind Regards, Carlos Velasco-Forero*

Many thanks for your detailed comment on the use of the Hanning window for the localisation of the 2D signal in space. Your concern that the shape of the window might alter the reproduction of the correct power spectrum does make a lot of sense. During our analysis, we tested the method with some of the most common window functions (namely the rectangular, Bartlett, Hanning, Hamming, Blackman and Kaiser windows) and found the Hanning to deliver the best results (most of the others were easily excluded since they produced strong discontinuities at the edges).

Your idea to use a top flat window to capture more of the rainfall field and hence improve on the above is really relevant. As you suggest, a dedicated study on the spectral properties of different window types and their impacts on the reproduction of the local spatial correlation structure of the rainfall field might bring some insights on choice of the optimal window function and help to reduce some of the artefacts that are currently associated with the method.

Following your Short Comment, we took the opportunity to add a small paragraph in our future perspective section that mentions your idea of a top flat window and more generally the relevance of investigating in more detail the choice of the window type.

[revised manuscript text omitted]